# Inference-Time Scaling for Joint Audio-Video Generation

**Jaemin Jung**                                                              *jjm5811@kaist.ac.kr*
*Korea Advanced Institute of Science and Technology*

**Kyeongha Rho**                                                            *khrho325@kaist.ac.kr*
*Korea Advanced Institute of Science and Technology*

**Inkyu Shin**                                                              *dlsrbgg33@gmail.com*
*Luma AI*

**Joon Son Chung**                                                          *joonson@kaist.ac.kr*
*Korea Advanced Institute of Science and Technology*

**Reviewed on OpenReview:** *https://openreview.net/forum?id=MHNFjjm5nO*

## Abstract

Joint audio-video generation aims to synthesize realistic audio-video pairs that are both semantically aligned with text prompts and precisely synchronized. While existing joint audio-video generation models often require substantial training resources to improve fidelity, Inference-Time Scaling (ITS) has recently emerged as a promising training-free alternative in single-modality domains. However, extending ITS from a single modality to multimodal domains is non-trivial, as it requires balancing multiple heterogeneous objectives. In this paper, we present the first comprehensive study of ITS for joint audio-video generation. We first demonstrate that a multi-verifier framework is essential to address the limitations of single-objective guidance, including asymmetric performance trade-offs and verifier hacking. Through systematic analysis, we then identify an optimal multi-verifier combination that yields balanced improvements across all quality dimensions. Finally, to effectively aggregate diverse reward signals, we propose Adaptive Reward Weighting (ARW), a novel test-time optimization algorithm. ARW treats reward aggregation as an online optimization problem, utilizing learnable parameters to calibrate reward variances without requiring prior knowledge of reward distributions, thereby ensuring robust multi-objective selection. Experimental results on VGGSound and JavisBench-mini benchmarks demonstrate that our framework significantly enhances semantic alignment, perceptual quality, and audio-visual synchronization of generated outputs. Synthesized samples and code are available on the project page: `https://jung-jaemin.github.io/ITS-AVGen-Proj`.

## 1 Introduction

Diffusion-based generative models (Song et al., 2021; Ho et al., 2020) have increasingly pushed the boundaries in AI-generated content (AIGC), significantly enhancing the fidelity of generated images (Rombach et al., 2022; Xie et al., 2025a), audios (Liu et al., 2023; Evans et al., 2025; Jung et al., 2025a), and videos (Ho et al., 2022; Chen et al., 2024). While earlier works primarily focused on the generation of a single-modality, recent research has shown growing interest in multimodal generation (Bao et al., 2023; Xu et al., 2023; Zhou et al., 2025a), in which multiple modalities are synthesized together coherently. Among these directions, joint audio-video generation (Ruan et al., 2023; Hayakawa et al., 2025; Liu et al., 2026) has emerged as a critical

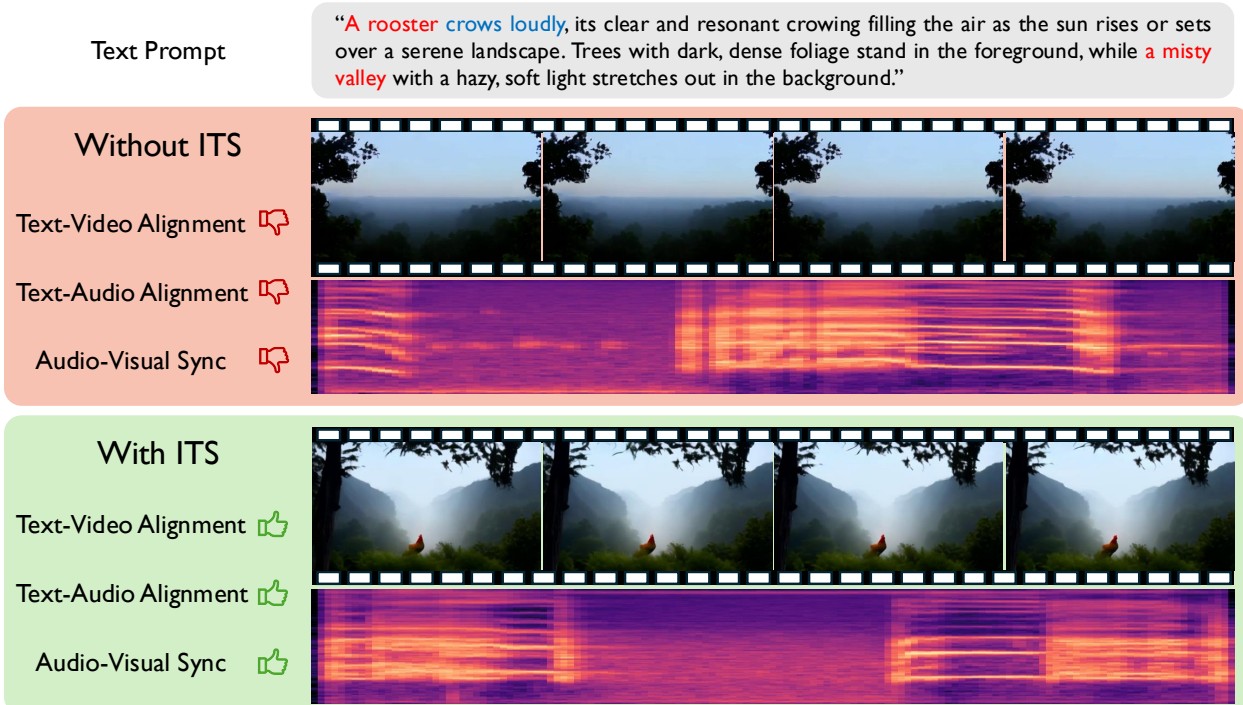

Figure 1: **Qualitative results with Inference-Time Scaling.** Compared to naive sampling, ITS yields audio–video pairs with superior semantic alignment with the text prompt and cross-modal synchronization. The video samples are available at the following link.

challenge. Since auditory and visual signals are tightly coupled in reality, generating them jointly is valuable for practical applications such as film production, game content creation, and AR/VR environments.

Joint audio-video generation aims to simultaneously synthesize auditory and visual streams that are temporally synchronized and semantically aligned. MM-Diffusion (Ruan et al., 2023) first introduced this task by proposing a unified framework to learn the joint distribution of audio and video latents. Following this, recent advancements (Liu et al., 2026; Hayakawa et al., 2025; Low et al., 2025) have applied text-conditioned diffusion models to joint audio-video generation, employing mechanisms like cross-attention to guide the simultaneous generation process with semantic cues from input text prompts. However, existing approaches still frequently fail to generate samples that faithfully match the text prompt. As a result, users are often forced to run inference repeatedly until satisfactory outputs are obtained, ultimately degrading the overall user experience. Mitigating this issue typically requires training larger models with additional data, which in turn demands substantial training time and computing resources (Kaplan et al., 2020; Hoffmann et al., 2022; Peebles & Xie, 2023).

To address the limitations of enormous training costs, Inference-Time Scaling (ITS) (Snell et al., 2025; Brown et al., 2024) has emerged as a promising alternative in single-modality generative tasks. It improves output quality without retraining the base generator by allocating more inference-time computation. In practice, it typically generates multiple candidate samples and selects outputs using external verifiers or reward signals. This paradigm is especially appealing for diffusion models (Ma et al., 2025; He et al., 2025a; Singhal et al., 2025) because their iterative sampling process is highly sensitive to stochasticity, where the choices of initial noise and intermediate trajectories can profoundly affect perceptual quality and prompt alignment of the final output. However, despite its success in single-modality settings, ITS has not been explored for multimodal domains, particularly for joint audio-video generation. Extending ITS from single-modality to joint audio–video generation is far from a trivial addition of an extra modality. It presents a fundamentally more complex challenge, as it demands the simultaneous optimization of multiple heterogeneous objectives. Specifically, achieving high-quality joint audio–video generation requires satisfying the following criteria

simultaneously: (1) semantic alignment between text prompts and generated modalities, (2) high perceptual quality of each generated modality, (3) semantic consistency between audio and video, and (4) precise audio–video synchronization.

In this paper, we take a first step toward extending ITS to joint audio–video generation. First, we experimentally demonstrate that a multi-verifier framework is essential to comprehensively take into account the key elements of joint audio–video generation. Single verifiers often fail to capture the multi-faceted nature of joint audio-video quality. More importantly, they are susceptible to verifier hacking, a failure mode where the search algorithm exploits verifier-specific biases to inflate a single metric. This results in 'hacked' outputs that achieve high scores but lack genuine improvements in perceptual quality or cross-modal coherence. Second, to address this limitation, we identify the optimal multi-verifier combination that improves performance across all four metrics required for high-quality generation. We begin by prioritizing text–video consistency, as it most directly influences user satisfaction in text-conditioned joint audio-video generation. After selecting an appropriate verifier for text-video consistency as the primary signal, we then incorporate complementary verifiers to consider other criteria. We find that audio-visual synchronization-based guidance is the most effective, yielding balanced improvements across all evaluation dimensions without performance trade-offs. Finally, to aggregate heterogeneous reward signals with different scales and distributions, we introduce Adaptive Reward Weighting (ARW), a test-time optimization approach that calibrates reward scales without requiring prior knowledge of reward distributions. ARW treats reward aggregation as an online optimization problem, assigning learnable calibration parameters to each reward type. By penalizing high-variance reward signals, ARW prevents any single reward from dominating the final aggregated score, ensuring a balanced contribution from all objectives. This directly addresses the limitations of conventional aggregation methods that depend on training-set statistics unavailable in real-world scenarios (Fig. 2), thereby enabling robust multi-objective selection.

Through extensive experiments on text-conditioned joint audio-video generation benchmarks (VGGSound test set (Chen et al., 2020) and JavisBench-mini (Liu et al., 2026)), we demonstrate that applying ITS with our proposed framework yields substantial improvements across all critical dimensions: semantic alignment, perceptual quality, and fine-grained synchronization. Results suggest that ITS is a practical and effective approach for advancing joint audio-video generation, provided that the complex interplay of multimodal objectives is managed through a deliberate multi-verifier design and a robust reward aggregation strategy.

Our main contributions can be summarized as follows:

- We present the first comprehensive study of ITS for joint audio-video generation. We identify the limitations of single-verifier guidance—verifier hacking and asymmetric performance trade-offs—and establish the necessity of a multi-verifier framework to ensure holistic quality improvements.

- We introduce an optimal multi-verifier combination that effectively balances heterogeneous objectives. By systematically evaluating various reward combinations, we demonstrate that integrating fine-grained synchronization guidance with text–video consistency offers the most robust performance, simultaneously enhancing semantic alignment and temporal synchronization.

- We propose Adaptive Reward Weighting, a novel test-time optimization algorithm for aggregating heterogeneous reward signals. This enables robust multi-objective selection without relying on prior knowledge of reward distributions or offline statistics.

## 2 Multi-Verifier Guidance for ITS in Joint Audio-Video Generation

### 2.1 Preliminary

**Joint Audio-Video Generation.** Joint audio-video generation aims to model the joint distribution of an audio-visual pair $x \triangleq (v, a)$. In this paper, we leverage diffusion models (Ho et al., 2020; Song et al., 2021) to synthesize both modalities together by reversing the gradual noising process. Specifically, for timestep

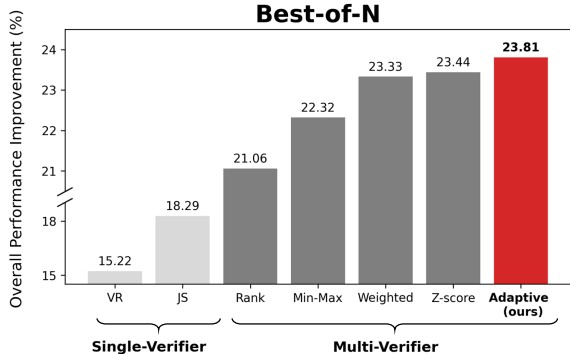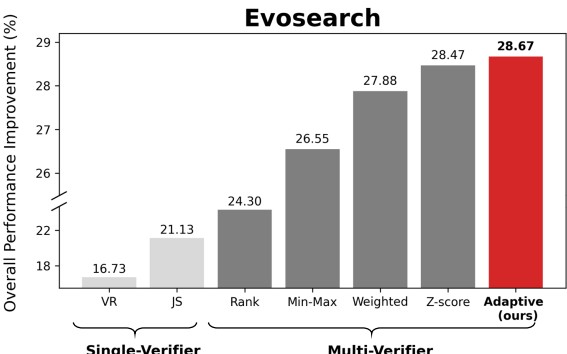

Figure 2: **Comparison of reward guidance types and aggregation methods.** The overall performance improvement is calculated by averaging the relative improvements of all evaluation metrics compared to the naive sampling (without ITS). This result is obtained with JavisDiT.

$t \in \{1, \ldots, T\}$, the forward process, in which the clean data $x_0$ is transformed into a noisy state $x_t$, is formulated as

$$x_t = \sqrt{\bar{\alpha}_t}x_0 + \sqrt{1 - \bar{\alpha}_t}\epsilon, \qquad \epsilon \sim \mathcal{N}(0, I), \qquad \bar{\alpha}_t \triangleq \prod_{\tau=1}^{t}(1 - \beta_\tau), \tag{1}$$

where $\{\beta_t\}_{t=1}^{T}$ denotes the noise schedule and $\epsilon$ represents standard Gaussian noise. Then, a conditional denoising network $\epsilon_\theta(x_t, s, t)$ is trained to predict the injected noise given the noisy state, the text prompt $s$, and the timestep. At inference time, the denoiser is used to generate new samples via a reverse-time Markov chain $p_\theta(x_{t-1} \mid x_t, s)$. Depending on the architecture, the denoiser can be implemented as a single joint network operating on the concatenated input $x_t$ (Liu et al., 2026), or as two separate unimodal diffusion models coupled via additional guidance mechanisms (Hayakawa et al., 2025).

**Inference-Time Scaling.** Inference-Time Scaling (ITS) for diffusion (Ma et al., 2025) or flow matching models (Kim et al., 2025a) refers to improving sample quality by spending additional compute on searching for better samples, without updating model parameters. Formally, given a pretrained conditional generator $p_\theta(x \mid s)$ and a verifier $V(x, s)$, the objective is to draw samples from a reward-tilted target distribution, defined as follows:

$$p^\star(x \mid s) \propto p_\theta(x \mid s) \exp\big(\lambda \cdot V(x, s)\big), \tag{2}$$

where $\lambda$ controls the strength of reward optimization relative to the output distribution of the pretrained generator. Direct sampling from $p^\star$ is generally intractable in high-dimensional latent spaces, so practical ITS methods approximate it with explicit search procedures guided by reward feedback.

The simplest ITS baseline is Best-of-$N$ sampling (Ma et al., 2025; Xie et al., 2025b), which generates $N$ independent candidates and returns the sample with the highest reward. While Best-of-$N$ sampling is widely applicable, it can be search-inefficient because it allocates full sampling compute to many candidates that are ultimately discarded. To address the inefficiency of Best-of-$N$ sampling, EvoSearch (He et al., 2025a) reframes ITS as an evolutionary search along the denoising trajectory. Unlike Best-of-$N$, which evaluates samples only after the final denoising step, EvoSearch performs intermediate evaluation and refinement at a predefined set of evolution steps. At each evolution step, only elite candidates with high rewards are selected and duplicated, and underperforming samples are replaced with mutated candidates generated by adding subtle noise to the elites' noise latents. This iterative cycle of evaluation, selection, and mutation ensures both population diversity and objective-aligned quality, providing more granular and effective guidance compared to standard post-hoc sampling.

Table 1: **Evaluation of JavisDiT with single-verifier and multi-verifier guidance on JavisBench-mini.** Text improvement and AV improvement are defined as the average performance gains over naive sampling for text-related and AV-related metrics, respectively. VR: VideoReward-TA, VQA: VQAScore, AVH: AVHScore, JS: JavisScore. TV-IB, TA-IB, and AV-IB are ImageBind similarity-based scores.

| Methods | Text-Consistency | | | | AV-Consistency | | AV-Sync | Improvement (%) | | |
|---|---|---|---|---|---|---|---|---|---|---|
| | VR ↑ | VQA ↑ | TV-IB ↑ | TA-IB ↑ | AV-IB ↑ | AVH-Score ↑ | JavisScore ↑ | Text ↑ | AV ↑ | Overall ↑ |
| Naive sampling | −0.478 | 0.852 | 0.275 | 0.146 | 0.209 | 0.188 | 0.161 | − | − | − |
| **Best-of-N** | | | | | | | | | | |
| Single (VR) | −0.040 | 0.892 | 0.283 | 0.148 | 0.213 | 0.192 | 0.164 | **25.15** | 1.97 | 15.22 |
| Single (JS) | −0.463 | 0.865 | 0.277 | 0.167 | 0.279 | 0.255 | 0.224 | 4.94 | **36.09** | 18.29 |
| Multi (VR + JS) | −0.106 | 0.889 | 0.282 | 0.161 | 0.252 | 0.231 | 0.201 | 23.75 | 22.76 | **23.33** |
| **Evosearch** | | | | | | | | | | |
| Single (VR) | 0.033 | 0.892 | 0.283 | 0.149 | 0.209 | 0.189 | 0.161 | **29.14** | 0.18 | 16.73 |
| Single (JS) | −0.501 | 0.841 | 0.272 | 0.174 | 0.296 | 0.273 | 0.240 | 3.00 | **45.30** | 21.13 |
| Multi (VR + JS) | −0.037 | 0.891 | 0.282 | 0.164 | 0.262 | 0.240 | 0.210 | 27.93 | 27.82 | **27.88** |

## 2.2 Is Single-Verifier Guidance Sufficient for ITS in Joint Audio-Video Generation?

In this section, we first investigate whether a single reward signal is sufficient for ITS in joint audio-video generation. To this end, we evaluate the performance of single-verifier guidance by employing two distinct verifiers: VideoReward-TA (VR) (Liu et al., 2025b), which targets text-video consistency, and JavisScore (JS) (Liu et al., 2026), which focuses on fine-grained audio-video synchronization.

As shown in Tables 1 and 2, our experiments reveal that single-verifier guidance effectively improves its intended evaluation metrics, yet fails to achieve a balanced improvement across all metrics. Specifically, relying solely on semantic guidance (VR) effectively enhances text-consistency metrics but yields only marginal or limited gains for audio-video alignment. Conversely, utilizing synchronization guidance (JS) alone substantially boosts audio-visual metrics; however, it provides limited improvement for semantic alignment.

These asymmetric trade-offs appear consistently across different search strategies, indicating that single-verifier settings are vulnerable to verifier hacking — a phenomenon where the inference-time search algorithm exploits blind spots of a specific reward rather than improving the genuine quality of the generated audio and video. In contrast, the multi-verifier framework (VR+JS) demonstrates robust and balanced improvements across all evaluation metrics. By combining complementary verifiers, this approach mitigates reward-specific bias, achieving simultaneous gains in text consistency and audio-video alignment without sacrificing either aspect. These findings underscore the indispensability of a multi-objective verification strategy for robust ITS in the multimodal domain.

## 2.3 What Is the Best Multi-Verifier Combination for ITS in Joint Audio-Video Generation?

In this section, we identify the optimal multi-verifier combination that consistently improves all metrics required for high-quality joint audio-video generation. Since text–video consistency is the most critical factor for user satisfaction in this domain, we adopt VideoReward-TA (VR), a video reward model trained on human preference data, as our primary verifier.

With VR fixed as our primary verifier to ensure a stable semantic anchor, we systematically evaluate complementary verifiers that target audio–visual alignment, which requires both cross-modal semantic coherence and fine-grained temporal synchronization. Specifically, we consider three auxiliary verifiers: (1) ImageBind-TA (TA-IB) (Girdhar et al., 2023), which encourages semantic coherence between the text prompt and the generated audio; (2) AVHScore (AVH) (Mao et al., 2024), which measures semantic consistency between audio and visual events; and (3) JavisScore (JS), which focuses on fine-grained audio–video synchronization.

Table 2: **Evaluation of MMDisCo with single-verifier and multi-verifier guidance on VGGSound test set.** Text improvement and AV improvement are defined as the average performance gains over naive sampling for text-related and AV-related metrics, respectively.

| Methods | Text-Consistency | | | | AV-Consistency | | AV-Sync | Improvement (%) | | |
|---|---|---|---|---|---|---|---|---|---|---|
| | VR ↑ | VQA ↑ | TV-IB ↑ | TA-IB ↑ | AV-IB ↑ | AVH-Score ↑ | JavisScore ↑ | Text ↑ | AV ↑ | Overall ↑ |
| Naive sampling | −1.104 | 0.597 | 0.300 | 0.179 | 0.189 | 0.182 | 0.160 | – | – | – |
| **Best-of-N** | | | | | | | | | | |
| Single (VR) | −0.440 | 0.665 | 0.316 | 0.183 | 0.201 | 0.196 | 0.176 | 19.77 | 8.01 | 14.73 |
| Single (JS) | −0.999 | 0.603 | 0.307 | 0.231 | 0.289 | 0.288 | 0.268 | 10.48 | **59.55** | **31.51** |
| Multi (VR + JS) | −0.513 | 0.665 | 0.317 | 0.210 | 0.248 | 0.244 | 0.225 | **21.98** | 35.31 | 27.69 |
| **Evosearch** | | | | | | | | | | |
| Single (VR) | −0.192 | 0.688 | 0.321 | 0.176 | 0.194 | 0.188 | 0.170 | 25.79 | 4.07 | 16.48 |
| Single (JS) | −0.973 | 0.622 | 0.307 | 0.249 | 0.320 | 0.324 | 0.304 | 14.38 | **79.11** | 42.12 |
| Multi (VR + JS) | −0.312 | 0.681 | 0.322 | 0.239 | 0.288 | 0.285 | 0.267 | **31.67** | 58.62 | **43.22** |

Table 3: **Comparison of multi-verifier combinations.** JavisDiT is evaluated on JavisBench-mini and MMDisCo is evaluated on VGGsound test set. All : VR + TA-IB + AVH + JS.

| Setup | Text-Consistency | | | | AV-Consistency | | AV-Sync | Improvement (%) | | |
|---|---|---|---|---|---|---|---|---|---|---|
| | VR ↑ | VQA ↑ | TV-IB ↑ | TA-IB ↑ | AV-IB ↑ | AVH ↑ | Javis ↑ | Text ↑ | AV ↑ | Overall ↑ |
| **JavisDiT** | | | | | | | | | | |
| Naive | −0.478 | 0.852 | 0.275 | 0.146 | 0.209 | 0.188 | 0.161 | – | – | – |
| VR + TA-IB | −0.138 | 0.886 | 0.279 | 0.162 | 0.227 | 0.206 | 0.178 | **21.88** | 9.58 | 16.61 |
| VR + AVH | −0.177 | 0.884 | 0.282 | 0.162 | 0.257 | 0.234 | 0.203 | 20.06 | 24.51 | 21.96 |
| VR + JS | −0.167 | 0.885 | 0.281 | 0.163 | 0.263 | 0.240 | 0.210 | 20.69 | **27.98** | **23.81** |
| All | −0.300 | 0.875 | 0.279 | 0.167 | 0.267 | 0.244 | 0.213 | 13.94 | 29.95 | 20.80 |
| **MMDisCo** | | | | | | | | | | |
| Naive | −1.104 | 0.597 | 0.300 | 0.179 | 0.189 | 0.182 | 0.160 | – | – | – |
| VR + TA-IB | −0.562 | 0.651 | 0.313 | 0.237 | 0.236 | 0.235 | 0.214 | **23.72** | 29.25 | 26.09 |
| VR + AVH | −0.600 | 0.653 | 0.314 | 0.222 | 0.266 | 0.262 | 0.241 | 20.93 | **45.11** | 31.29 |
| VR + JS | −0.574 | 0.657 | 0.319 | 0.216 | 0.264 | 0.258 | 0.240 | 21.27 | 43.81 | 30.93 |
| All | −0.799 | 0.630 | 0.311 | 0.240 | 0.282 | 0.280 | 0.260 | 17.73 | 55.19 | **33.78** |

As shown in Table 3, the synchronization-based guidance (VR+JS) simultaneously improves text consistency and audio-visual alignment in both JavisDiT and MMDisCo, achieving the most balanced overall performance improvement. By comparison, the text–audio alignment-based guidance (VR+TA-IB) offers only modest improvements in audio-visual semantic alignment. The audio-video semantic alignment-based guidance (VR+AVH) substantially boosts both text consistency and audio-video alignment, but still falls short of the performance level attained by the synchronization-based guidance (VR+JS). We attribute this to the fact that audio-visual synchronization represents temporally fine-grained audio-visual semantic consistency, i.e., a unified notion that encompasses both temporal alignment (when events happen) and semantic correspondence (what events happen) across modalities. TA-IB and AVH primarily rely on global audio embeddings, which makes it difficult to reliably evaluate temporally fine-grained semantic agreement. In contrast, JS leverages segment-wise audio representations to assess event consistency at a much finer temporal granularity. Therefore, it provides a powerful complementary signal to VR that improves alignment without undermining the text-video semantic anchor.

We further compare our approach with a four-verifier setting to test whether simply adding more verifiers consistently yields higher-quality outputs. However, using more verifiers does not necessarily translate into better overall performance. When all four verifiers are jointly applied, the resulting performance gains become skewed toward audio-video alignment rather than remaining balanced across objectives. A plausible explanation is that adding multiple verifiers implicitly overweights audio-related objectives, leading to a bias

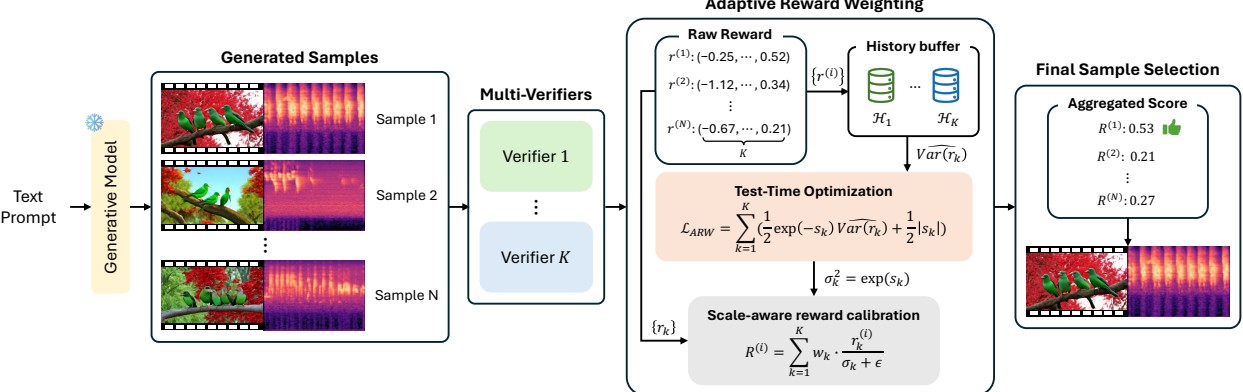

Figure 3: **Overview of Adaptive Reward Weighting.** Given audio-video samples generated by the pre-trained generative model, they are evaluated by multiple verifiers, with their raw rewards stored in a history buffer. Then, the aggregated score is computed via a weighted sum of these rewards using reward-specific learnable calibration parameters, and calibration parameters are updated through test-time optimization. The final sample is selected based on the resulting aggregated score.

that prioritizes audio-video alignment at the expense of prompt-faithful video generation. Furthermore, the inclusion of TA-IB and AVH introduces a degree of semantic redundancy. Because JS already captures these relationships at a more granular, frame-wise level, the additional verifiers are more likely to act as optimization noise rather than providing constructive guidance. Moreover, additional verifiers increase computational overhead, making ITS substantially slower and less practical.

In summary, the experimental results suggest that a multi-verifier guidance combining text-video consistency and audio-visual synchronization as reward signals offers the optimal accuracy-efficiency trade-off for ITS in joint audio-video generation.

## 3 Adaptive Reward Weighting

When optimizing multimodal generative outputs using multiple reward signals, a critical design choice is how to aggregate heterogeneous rewards into a single scalar value for sample selection. We first review existing aggregation methods and then introduce our proposed adaptive reward weighting.

### 3.1 Conventional Methods

Prior works have explored several strategies for combining multiple reward signals. **Weighted Sum** (Kim et al., 2025b) computes a linear combination of rewards $R = \sum_k w_k r_k$ with predefined weights $w_k$, which typically requires extensive hyperparameter tuning and may fail to generalize across diverse prompts. **Rank-based aggregation** (Liu et al., 2025a; Jin et al., 2025) converts each reward into its percentile rank within the candidate set and sums these ranks, avoiding scale mismatch but discarding magnitude information. **Min–Max normalization** rescales each reward to $[0, 1]$ using $\hat{r}_k = (r_k - r_k^{\min})/(r_k^{\max} - r_k^{\min})$, where $r_k^{\min}$ and $r_k^{\max}$ are computed over the candidate set; this approach is sensitive to outliers. **Z-score normalization** (Jung et al., 2025b) standardizes rewards via $\hat{r}_k = (r_k - \mu_k)/\sigma_k$, which typically relies on precomputed population statistics and may not generalize under distribution shifts. These methods share a common limitation: they rely on static aggregation rules that cannot adapt to the varying reward characteristics observed during inference.

### 3.2 Adaptive Reward Weighting

To overcome the limitations of static aggregation, we propose **Adaptive Reward Weighting (ARW)**, a test-time optimization method that calibrates heterogeneous reward signals and aggregates them into a single scalar score for inference-time sample selection. The key challenge is that different verifiers output

rewards with disparate scales and variances, and these distributions can vary depending on the prompts and candidate sets. A naive summation of these rewards allows the reward with a larger variance to dominate the aggregated score, thereby suppressing the influence of other rewards.

**Scale-aware Reward Calibration.** Let $\{r_k^{(i)}\}_{k=1}^K$ denote the reward values assigned to candidate $i$ by $K$ verifiers. ARW assigns each reward a learnable *calibration parameter* $\sigma_k > 0$ to adjust reward values at test time. Then the aggregated score is computed by a weighted sum of rewards normalized by calibration parameters as follows:

$$R^{(i)} = \sum_{k=1}^K w_k \cdot \frac{r_k^{(i)}}{\sigma_k + \epsilon}, \tag{3}$$

where $w_k$ are optional preference weights that can be adjusted to steer guidance toward a specific reward signal, and $\epsilon$ is a small constant for numerical stability. This normalization standardizes the variance of each reward distribution, preventing any single objective from dominating the aggregated score due to scale disparities. Notably, we perform scale-only normalization without mean subtraction, as our primary goal is to equalize the relative fluctuations across heterogeneous rewards. Since our selection relies on the ranking of candidates, the process is invariant to mean shifts, making mean subtraction redundant.

**Learning Calibration Scales at Test Time.** Rather than setting $\sigma_k$ manually, ARW learns them from the reward statistics observed during inference. For each verifier $V_k$, we maintain a history buffer $\mathcal{H}_k = \{r_k^{(j)}\}_{j=1}^{|\mathcal{H}_k|}$ that accumulates reward values across multiple prompts and successive generation steps. To ensure $\sigma_k > 0$ during optimization, we parameterize the scale as $\sigma_k^2 = \exp(s_k)$ with a learnable log-scale parameter $s_k$. Using this history, we estimate the empirical variance as follows:

$$\widehat{\text{Var}}(r_k) = \frac{1}{|\mathcal{H}_k|} \sum_{j=1}^{|\mathcal{H}_k|} \left(r_k^{(j)} - \bar{r}_k\right)^2, \qquad \bar{r}_k = \frac{1}{|\mathcal{H}_k|} \sum_{j=1}^{|\mathcal{H}_k|} r_k^{(j)}. \tag{4}$$

We then update $\{s_k\}$ by minimizing the following objective adapted from uncertainty-based multi-task learning (Kendall et al., 2018):

$$\mathcal{L}_{ARW} = \sum_{k=1}^K \left(\frac{1}{2} \exp(-s_k) \widehat{\text{Var}}(r_k) + \frac{1}{2}|s_k|\right), \tag{5}$$

where the first term increases $s_k$ in response to high reward variance, reducing its relative influence on the aggregated score, while the second term regularizes the scale to prevent $s_k$ from diverging. In practice, we perform only a small number of gradient steps with a lightweight optimizer, making the computational overhead caused by this optimization process negligible relative to diffusion sampling and reward evaluation.

## 4 Experiments

### 4.1 Experiment Setup

**Evaluation Benchmarks and Models.** We evaluate our proposed Adaptive Reward Weighting (ARW) and other reward aggregation methods on two benchmarks: JavisBench-mini and the VGGSound (Chen et al., 2020) test set. Experiments are conducted with two joint audio–video generation models, JavisDiT(Liu et al., 2026) and MMDisCo(Hayakawa et al., 2025). JavisDiT is a flow-matching model that generates 4-second videos at 240p resolution, while MMDisCo is a discriminator-guided cooperative diffusion model that generates 2-second videos at $256 \times 256$ resolution. Additional model details are provided in Appendix B.1.

**Metrics.** We report four categories of metrics that comprehensively evaluate generated audio and video from various aspects: (1) Alignment between the text prompt and the generated modalities is measured using VideoReward-TA (Liu et al., 2025b), VQAScore (Lin et al., 2024), and ImageBind (Girdhar et al., 2023) scores computed for text–video (TV-IB) and text–audio (TA-IB). (2) Audio–video semantic alignment is measured by ImageBind audio–video similarity (AV-IB) and AVHScore (Mao et al., 2024). (3) Audio–video

Table 4: **Performance comparison of aggregation methods on the JavisBench-mini.** Text improvement and AV improvement are defined as the average performance gains over naive sampling for text-related and AV-related metrics, respectively. * denotes optimization targets.

| Aggregation | Text-Consistency | | | | AV-Consistency | | AV-Sync | Improvement (%) | | |
| --- | --- | --- | --- | --- | --- | --- | --- | --- | --- | --- |
| | VR* ↑ | VQA ↑ | TV-IB ↑ | TA-IB ↑ | AV-IB ↑ | AVH-Score ↑ | JavisScore* ↑ | Text ↑ | AV ↑ | Overall ↑ |
| Naive sampling | −0.478 | 0.852 | 0.275 | 0.146 | 0.209 | 0.188 | 0.161 | – | – | – |
| **Best-of-N** | | | | | | | | | | |
| Rank | −0.193 | 0.882 | 0.281 | 0.160 | 0.256 | 0.233 | 0.203 | 18.73 | 24.17 | 21.06 |
| Min-Max | −0.170 | 0.885 | 0.281 | 0.162 | 0.257 | 0.234 | 0.205 | 20.36 | 24.92 | 22.32 |
| Weighted | −0.106 | 0.889 | 0.282 | 0.161 | 0.252 | 0.231 | 0.201 | **23.75** | 22.76 | 23.33 |
| Z-score | −0.213 | 0.883 | 0.281 | 0.164 | 0.267 | 0.244 | 0.214 | 18.40 | **30.15** | 23.44 |
| ARW (Ours) | −0.167 | 0.885 | 0.281 | 0.163 | 0.263 | 0.240 | 0.210 | 20.69 | 27.98 | **23.81** |
| **Evosearch** | | | | | | | | | | |
| Rank | −0.167 | 0.883 | 0.281 | 0.165 | 0.264 | 0.241 | 0.212 | 20.97 | 28.73 | 24.30 |
| Min-Max | −0.141 | 0.884 | 0.282 | 0.166 | 0.270 | 0.247 | 0.217 | 22.63 | 31.78 | 26.55 |
| Weighted | −0.037 | 0.891 | 0.282 | 0.164 | 0.262 | 0.240 | 0.210 | **27.93** | 27.82 | 27.88 |
| Z-score | −0.157 | 0.884 | 0.280 | 0.167 | 0.280 | 0.258 | 0.227 | 21.78 | **37.40** | 28.47 |
| ARW (Ours) | −0.134 | 0.885 | 0.281 | 0.166 | 0.278 | 0.256 | 0.225 | 22.93 | 36.31 | **28.67** |

Table 5: **Performance comparison of aggregation methods on the VGGSound test set.** Text improvement and AV improvement are defined as the average performance gains over naive sampling for text-related and AV-related metrics, respectively. * denotes optimization targets.

| Methods | Text-Consistency | | | | AV-Consistency | | AV-Sync | Improvement (%) | | |
| --- | --- | --- | --- | --- | --- | --- | --- | --- | --- | --- |
| | VR* ↑ | VQA ↑ | TV-IB ↑ | TA-IB ↑ | AV-IB ↑ | AVH-Score ↑ | JavisScore* ↑ | Text ↑ | AV ↑ | Overall ↑ |
| Naive sampling | −1.104 | 0.597 | 0.300 | 0.179 | 0.189 | 0.182 | 0.160 | – | – | – |
| **Best-of-N** | | | | | | | | | | |
| Rank | −0.636 | 0.653 | 0.315 | 0.220 | 0.260 | 0.257 | 0.237 | 19.92 | 42.30 | 29.51 |
| Min-Max | −0.595 | 0.649 | 0.316 | 0.219 | 0.260 | 0.256 | 0.237 | 20.63 | 42.12 | 29.84 |
| Weighted | −0.513 | 0.665 | 0.317 | 0.210 | 0.248 | 0.244 | 0.225 | **21.98** | 35.31 | 27.69 |
| Z-score | −0.629 | 0.647 | 0.315 | 0.223 | 0.271 | 0.266 | 0.247 | 20.25 | **47.97** | **32.13** |
| ARW (Ours) | −0.591 | 0.653 | 0.315 | 0.221 | 0.264 | 0.259 | 0.240 | 21.08 | 44.00 | 30.90 |
| **Evosearch** | | | | | | | | | | |
| Rank | −0.490 | 0.670 | 0.320 | 0.226 | 0.275 | 0.271 | 0.253 | 25.20 | 50.84 | 36.19 |
| Min-Max | −0.455 | 0.682 | 0.319 | 0.227 | 0.278 | 0.273 | 0.256 | 26.55 | 52.36 | 37.61 |
| Weighted | −0.312 | 0.681 | 0.322 | 0.239 | 0.288 | 0.285 | 0.267 | **31.67** | 58.62 | 43.22 |
| Z-score | −0.452 | 0.671 | 0.318 | 0.248 | 0.309 | 0.306 | 0.288 | 29.00 | **70.54** | **46.80** |
| ARW (Ours) | −0.414 | 0.672 | 0.320 | 0.247 | 0.306 | 0.301 | 0.284 | 29.93 | 68.26 | 46.36 |

synchrony is quantified using JavisScore. (4) Video perceptual quality is evaluated using the VBench metric suite (Huang et al., 2024), following the evaluation setting of VideoScore2 (He et al., 2025b).

**Implementation Details.** For ITS, we set the number of generated samples per prompt to 5 for JavisDiT and 10 for MMDisCo. To update the balancing weights online in ARW, we use the Adam optimizer with a learning rate of 0.05. At each generation step, we run 50 optimization iterations to minimize the adaptive reward objective using accumulated reward statistics. Detailed ITS settings are provided in Appendix B.1. All experiments are conducted on a single NVIDIA RTX 4090 GPU.

## 4.2 Main Result

**Effectiveness of Adaptive Reward Weighting.** We demonstrate the effectiveness of our proposed ARW strategy by comparing it against standard aggregation methods across both datasets. As shown

Table 6: **Quantitative evaluation of video quality.** Video quality is evaluated using five metrics from VBench. We report scores obtained using the Best-of-N algorithm for both JavisDiT and MMDisCo.

| Setup | Video Quality | | | | | | Avg ↑ |
|---|---|---|---|---|---|---|---|
| | Subject ↑ | Motion ↑ | Background ↑ | Aesthetic ↑ | Image Quality ↑ | | Avg ↑ |
| **JavisDiT** | | | | | | | |
| Naive | 96.12 | 99.05 | 96.70 | 46.34 | 60.06 | | 79.65 |
| Single (VR) | 96.36 | 99.09 | 96.81 | 47.26 | 60.42 | | 79.99 |
| Single (JS) | 96.43 | 99.08 | 96.87 | 46.70 | 60.29 | | 79.87 |
| Z-score (VR + JS) | 96.41 | 99.08 | 96.86 | 47.14 | 60.38 | | 79.97 |
| ARW (VR + JS) | 96.42 | 99.08 | 96.87 | 47.19 | 60.46 | | **80.00** |
| **MMDisCo** | | | | | | | |
| Naive | 85.34 | 89.41 | 92.60 | 45.21 | 51.00 | | 72.71 |
| Single (VR) | 87.68 | 90.76 | 93.51 | 47.67 | 51.66 | | 74.26 |
| Single (JS) | 87.29 | 90.56 | 93.48 | 46.11 | 51.98 | | 73.88 |
| Z-score (VR + JS) | 88.24 | 90.97 | 93.82 | 47.34 | 52.77 | | 74.63 |
| ARW (VR + JS) | 88.25 | 90.98 | 93.78 | 47.54 | 52.96 | | **74.70** |

(a) JavisDiT

(b) MMDisCo

Figure 4: **Inference-time scaling curves across the number of samples.** Across both Best-of-$N$ and EvoSearch, the Z-score and ARW aggregation methods exhibit consistent performance improvement as the sample size increases. In both (a) and (b), the left and right panels show VR and JS scores, respectively.

in Table 4, ARW consistently achieves the highest overall improvement on JavisBench-mini, outperforming existing methods in both Best-of-$N$ and EvoSearch settings. Notably, conventional methods often suffer from a performance trade-off; for instance, the weighted sum excels in text consistency but lags in audio-video alignment, whereas the Z-score normalization tends to bias toward audio-video alignment. In contrast, ARW demonstrates superior robustness, effectively balancing these conflicting objectives to yield substantial gains in both text and audio-video metrics simultaneously. Crucially, ARW achieves this performance without requiring hyperparameter tuning or prior statistics of reward distributions.

This balance is particularly critical for an iterative search algorithm. The performance gap between Best-of-$N$ and EvoSearch across both datasets suggests that advanced search algorithms benefit significantly from robust reward aggregation. Since EvoSearch repeatedly updates the candidate population based on aggregated rewards, it is far more sensitive to scale mismatches and noisy signals than one-shot selection. In such scenarios, naive aggregation schemes can provide poorly calibrated guidance, thereby failing to steer the evolutionary process effectively. By contrast, ARW produces a stable, calibrated signal that enables EvoSearch to navigate the generation space more efficiently, yielding larger gains in challenging multimodal generation tasks. Furthermore, results on the VGGSound test set (Table 5) confirm the generalization capability of ARW, where it maintains competitive performance comparable to the Z-score normalization while avoiding extreme variances across modalities.

**Video Perceptual Quality.** As shown in Table 6, applying ITS consistently improves video quality compared to the naive sampling baseline. Using either VideoReward-TA (VR) or JavisScore (JS) individually yields higher perceptual scores than the unguided baseline. Building on this, aggregating these signals leads to further performance gains. Specifically, when VR and JS are combined via a scale-calibrated aggregation

Table 7: **Ablation on text-video consistency verifiers.** We compare the effectiveness of VQAScore and VideoReward-TA under the two ITS algorithms on MMDisCo. Video improvement is defined as the average performance gains over naive sampling for video quality metrics.

| Methods | Model | Text-Consistency | | | Video Quality | Improvement (%) | | |
| --- | --- | --- | --- | --- | --- | --- | --- | --- |
| | | VR ↑ | VQA ↑ | TV-IB ↑ | VBench ↑ | Text ↑ | Video ↑ | Overall ↑ |
| Naive sampling | - | -1.104 | 0.597 | 0.300 | 72.71 | – | – | – |
| Best-of-N | VQA | -0.737 | 0.715 | 0.313 | 74.79 | 19.11 | 2.86 | 15.05 |
| | VR | -0.440 | 0.665 | 0.316 | 74.26 | 25.62 | 2.13 | **19.75** |
| EvoSearch | VQA | -0.708 | 0.740 | 0.320 | 75.12 | 22.16 | 3.31 | 17.45 |
| | VR | -0.192 | 0.688 | 0.321 | 74.69 | 34.95 | 2.72 | **26.90** |

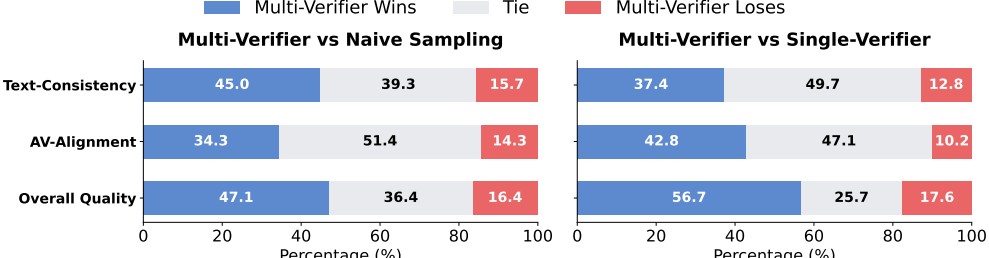

Figure 5: **Human evaluation results.**

method (Z-score or ARW), the overall quality surpasses that of single-reward guidance. Notably, our ARW strategy achieves the highest average quality scores on both JavisDiT and MMDisCo, underscoring its ability to optimally leverage complementary signals. We attribute the effectiveness of JS, originally designed for audio-visual synchronization, to its role as a proxy for temporal coherence. Specifically, JS enhances overall video quality by penalizing semantically correct but temporally misaligned samples, as demonstrated in the video available at this link.

**Impact of Inference-Time Scaling.** To evaluate the scalability of the proposed methods, we apply Z-score normalization and ARW to two ITS strategies while gradually increasing the samples. As illustrated in Fig. 4, we observe a consistent monotonic improvement in both VR and JS metrics across JavisDiT and MMDisCo as the computational budget increases. This confirms that our multi-verifier guidance effectively steers the generation process toward coherent and natural outputs, validating the fundamental premise that scaling inference-time computation leads to superior output quality in joint audio-video generation.

## 4.3 Ablation Studies

**Comparison of Verifiers for Text–Video Consistency.** We conduct an ablation study on MMDisCo to determine the more effective verifier between VQAScore (VQA) and VR under identical ITS pipelines, as shown in Table 7. First, regarding generation quality, both verifiers yield clear improvements in video perceptual quality over naive sampling. This suggests that reward-guided selection itself enhances visual fidelity, separately from the specific text consistency signal. However, VR demonstrates superior generalization in text–video alignment; it consistently achieves higher TV-IB scores for both Best-of-$N$ and EvoSearch. Since our primary objective is text consistency, VR is the more suitable choice. Second, regarding practicality, VR is significantly more efficient for ITS. While VQA typically requires a massive backbone (e.g., CLIP-FlanT5-11B (Lin et al., 2024)), VR operates effectively with a much smaller backbone (e.g., Qwen2-VL-2B (Wang et al., 2024b)). This efficiency is critical, as the high per-candidate evaluation cost of VQA becomes a severe bottleneck when processing thousands of candidates during the search process.

**Human Evaluation.** To validate the alignment of our method with human preferences, we conduct a comprehensive human evaluation study along three dimensions: text consistency, audio–video alignment,

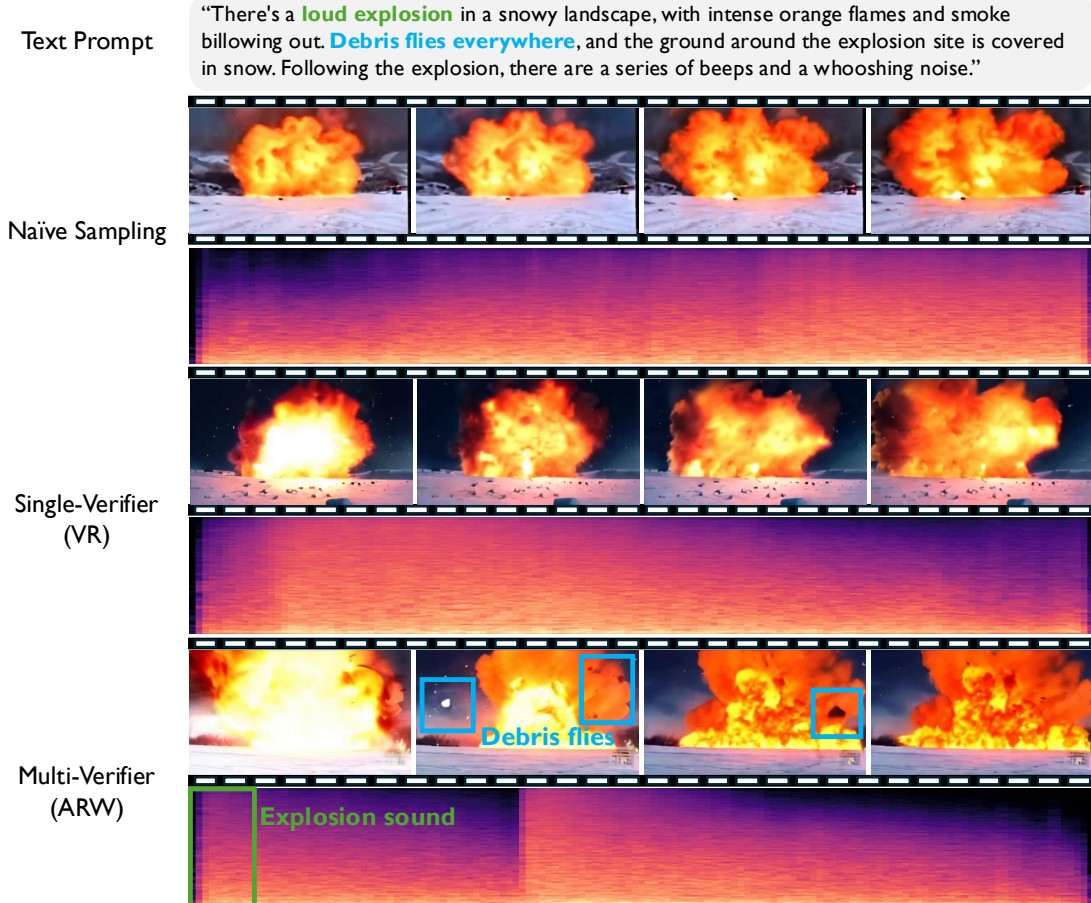

Figure 6: **Qualitative comparison of generated samples.** We compare the outputs of naive sampling, single-verifier (VR-guidance), and our multi-verifier (ARW) given a complex text prompt on JavisDiT. The video samples are available at the following link.

and overall quality. We recruited 16 participants, with each participant evaluating 20 sample sets to compare videos generated by naive sampling, single-verifier guidance using VR, and our multi-verifier guidance. As shown in Fig. 5, our multi-verifier guidance achieves higher win rates than both naive sampling and single-verifier guidance across all three criteria. Notably, the multi-verifier approach demonstrates a significant advantage in overall quality compared to the single-verifier baseline. This aligns with our analysis in Table 6, where JS was shown to improve fidelity by filtering out semantically consistent but temporally misaligned samples. These subjective findings further corroborate the objective evaluations reported in Table 1, solidly demonstrating the effectiveness of our multi-verifier framework.

## 4.4 Qualitative Results

In Fig. 6, we qualitatively compare naive sampling, single-verifier guidance (VR), and our multi-verifier method (ARW) given a complex text prompt. While naive sampling and single-verifier guidance often miss fine-grained details, our method successfully generates the specified visual element ("Debris flies," highlighted in blue) and synchronizes it with the corresponding audio event ("Explosion sound," highlighted in green). These results indicate that our multi-verifier framework achieves stronger text–audio–video alignment, capturing fine-grained semantic cues at multiple granularities that single-verifier methods often fail to synthesize. The video samples are available at the following link. Additional qualitative results for JavisDiT and MMDisCo are provided in Appendix E.

## 5 Limitations

Despite the consistent improvements enabled by ITS, several limitations remain to be addressed. First, current publicly available joint audio–video generation models still exhibit limited base fidelity. Therefore, even with advanced ITS methods, outputs may occasionally fail to fully satisfy the prompt, bounding the potential improvements. Second, the high memory footprint required for high-dimensional multimodal data significantly constrains the search budget, limiting the number of candidates that can be evaluated in parallel. Finally, ITS incurs substantial computational overhead due to the need for generating and scoring multiple candidates. While efficient inference methods have been actively studied in video generation (Xie et al., 2025a;b; Lv et al., 2025), it remains underexplored for joint audio–video generation. Future work should therefore focus on enhancing the fidelity of pretrained models and developing resource-efficient search algorithms to make scalable ITS practical for real-world applications.

## 6 Conclusion

In this paper, we present the first comprehensive study on Inference-Time Scaling (ITS) for joint audio-video generation. We demonstrate that relying on single-verifier guidance leads to asymmetric performance trade-offs and makes the system vulnerable to verifier hacking. To address these challenges, we identify that a multi-verifier framework combining text-video consistency with fine-grained audio-visual synchronization is essential for overall quality improvement. Furthermore, to effectively aggregate these heterogeneous signals, we propose Adaptive Reward Weighting (ARW), a novel test-time optimization algorithm. By calibrating reward variances via online optimization, ARW ensures robust multi-objective selection without requiring prior knowledge of reward distributions. Extensive experimental results on JavisBench-mini and VGGSound demonstrate that our approach significantly outperforms existing reward aggregation methods, validating ITS as a promising and practical direction for advancing multimodal generation.

## 7 Acknowledgment

This work was supported by IITP grants funded by the Korean government (MSIT, RS-2025-02263169, Detection and Prediction of Emerging and Undiscovered Voice Phishing).

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

## Appendix

In the appendix, we provide additional information as listed below:

- Appendix A provides the related works.

- Appendix B provides the experimental setup and computation.

- Appendix C provides the implementation details of Adative Reward Weighting.

- Appendix D provides extensive ablation study on the proposed methods.

- Appendix E provides qualitative results on JavisDiT and MMDisCo.

- Appendix F provides the licenses of the assets used.

- Appendix G discusses broader impact.

- Appendix H discusses failure cases.

- Appendix I discusses future work.

## A    Related Works

**Audio-Video Generation.**  Audio–video generation has recently emerged as a compelling direction in multimodal AI-generated content (AIGC), aiming to synthesize perceptually coherent and temporally synchronized video and audio together. Prior work on audio–video generation broadly falls into two categories. The first is cascaded generation, which follows a sequential pipeline where one modality is generated first to serve as a condition for the other (Luo et al., 2023; Zhang et al., 2025; Ren et al., 2025; Polyak et al., 2024; Jeong et al., 2023). While these pipelines can leverage the powerful ability of specialized single-modality generators, they often suffer from error propagation across stages and limited cross-modal interaction, making it difficult to achieve fine-grained synchronization beyond coarse semantic consistency (Luo et al., 2023; Ren et al., 2025).

The second line of research pursues end-to-end joint audio-video generation, by modeling the joint distribution of both modalities within a unified diffusion framework. Early works, primarily adopting U-Net-based Ronneberger et al. (2015) diffusion models, incorporate cross-modal interaction blocks or shared latent representations to couple the denoising trajectories (Ruan et al., 2023; Xing et al., 2024; Sun et al., 2024). MMDisCo (Hayakawa et al., 2025) introduces a discriminator-guided cooperative diffusion strategy, enforcing audio-visual consistency through iterative guidance while maintaining the specialized generative power of modality-specific experts.

More recently, the community has increasingly adopted Diffusion Transformer (DiT) architectures (Peebles & Xie, 2023) due to their scalability and strong generation capability. Representative DiT-based designs emphasize efficient integration of modalities: AV-DiT (Wang et al., 2024a) utilizes modality-specific adapters within a shared backbone, UniForm (Zhao et al., 2025) adopts token-level concatenation, and SyncFlow (Liu et al., 2024) employs directed cross-modal conditioning to encourage temporal alignment. Building on these foundations, JavisDiT (Liu et al., 2026) further prioritizes fine-grained spatio-temporal synchronization by injecting hierarchical priors directly into DiT blocks. Despite these advances in training strategies, the ability to adaptively refine generation at inference time without costly retraining remains underexplored.

**Inference-Time Scaling for Diffusion Models.**  Inspired by the success of Large Language Models (LLMs) in improving reasoning through additional inference-time computation (Snell et al., 2025; Wu et al., 2025; Liu et al., 2025c), Inference-Time Scaling (ITS) has recently emerged as a promising direction for diffusion models. Beyond the straightforward approach of increasing denoising steps, recent literature suggests that ITS can be framed as allocating compute to facilitate better decision-making during the generation process (Ma et al., 2025). A key finidng in this domain is that the choice of the initial noise seed significantly

dictates final output quality, motivating efforts to identify or optimize *golden noise* seeds that yield superior results (Zhou et al., 2025b).

Current ITS methodologies typically allocate additional compute in two ways: (i) refinement of the initialization, such as injecting temporal correlation via noise warping (Chang et al., 2024) or iteratively optimizing latent frequency components (Wu et al., 2024; Yuan et al., 2025); and (ii) search-based selection, which explores multiple candidates to identify the optimal output according to a specific scoring function. For instance, the latent beam search (Oshima et al., 2025) maintains a set of promising latent candidates at each denoising step and prunes lower-quality paths to concentrate compute on high-reward trajectories. On the other hand, the evolutionary search (He et al., 2025a) reformulates the sampling process as an evolutionary optimization problem, applying selection and mutation mechanisms to intermediate denoising states to iteratively guide the population toward higher-reward regions while preserving sample diversity.

However, a critical limitation of existing ITS frameworks is that they have only been applied to unimodal generation. Although current verifiers and selection criteria effectively evaluate unimodal fidelity, they are not designed to capture the complex cross-modal consistency—such as cross-modal semantic alignment and temporal synchronization—essential for multimodal generation. To the best of our knowledge, this work presents the first ITS framework specifically tailored to joint audio-video generation, addressing the unique challenge of optimizing both quality of individual modality and inter-modal alignment at inference time.

**Multi-Objective Optimization.** Balancing multiple verifier scores for multimodal generation naturally frames our inference-time search as a Multi-Objective Optimization (MOO) problem. Conventional MOO approaches are predominantly designed for the training phase. For instance, Pareto-based methods (Sener & Koltun, 2018; Lin et al., 2019) aim to find gradient update directions that achieve Pareto-optimal trade-offs among tasks. Similarly, uncertainty-aware multi-task learning (Kendall et al., 2018) dynamically weights loss functions during training based on task-specific uncertainty. Despite their effectiveness, these approaches require access to the model's internal parameters and involve computationally heavy gradient calculations over a training dataset.

In contrast, our proposed ARW introduces an inference-time paradigm that does not require updating the parameters of the base generators. ARW conceptually adapts the intuitive principle of uncertainty-based weighting (Kendall et al., 2018), but uniquely repurposes it for test-time search over frozen generative models. By dynamically calibrating the scales of heterogeneous reward signals using a few learnable parameters, ARW resolves multi-objective conflicts on the fly. This provides a unique advantage as a highly scalable, training-free multi-objective search algorithm tailored for the computational constraints of ITS.

# B  Experimental Setup and Computational Cost

## B.1  Model and Inference Settings

Table 8 summarizes the experimental configurations for JavisDiT and MMDisCo. The table details the model architectures, inference settings, and the hyperparameters used for EvoSearch.

## B.2  Compute-Performance Trade-off.

Table 9 summarizes the computational cost of the two ITS strategies, measured by wall-clock time and the number of function evaluations (NFE), for both JavisDiT and MMDisCo under multi-verifier guidance. Compared to naive sampling, Best-of-$N$ and EvoSearch require substantially more computation, leading to increased runtime. However, this additional inference cost translates into clear performance gains: both methods consistently improve text, audio-video, and overall scores over the baseline, with EvoSearch achieving the largest improvements at the highest computational cost. Beyond the choice of search strategy, we also examine the overhead of the aggregation method. Across all settings, ARW increases wall-clock time by only 0.06–0.08% compared to weighted-sum aggregation. This is because ARW performs only a small number of gradient updates with a lightweight optimizer to calibrate reward scales online; in practice, the resulting overhead is negligible relative to the total time spent on diffusion sampling and reward evaluation.

Table 8: **Model configurations and EvoSearch hyperparameters.** The table summarizes the architecture details and inference settings for JavisDiT and MMDisCo.

| Component | JavisDiT | MMDisCo |
|---|---|---|
| *Model Architecture* | | |
| Model type | JavisDiT-v0.1 | VideoCrafter2 + Auffusion |
| Generation paradigm | Flow-matching | Discriminator-guided cooperative diffusion |
| Parameters | 3.14B | 132M |
| Output spec | 4s video @ 240p | 2s video @ $256 \times 256$ |
| Video VAE | OpenSoraVAE V1.2 | AutoencoderKL (VideoCrafter) |
| Audio VAE | AudioLDM2 | AutoencoderKL (Auffusion) |
| Text encoder | T5-v1.1-XXL | FrozenOpenCLIPEmbedder |
| Vocoder | HiFi-GAN | HiFi-GAN |
| *Sampling Configuration* | | |
| Scheduler | Rectified Flow | DDPM |
| Use timestep transform | True | False |
| Sampling steps | 30 | 50 |
| CFG scale | 7.0 | 8.0 |
| *EvoSearch Configuration* | | |
| Evolution schedule | Steps [0, 10] | Steps [0, 20] |
| Population size | [5, 5, 5] | [10, 10, 10] |
| Elite size | 2 | 2 |
| Mutation rate | 0.2 | 0.2 |
| Tournament ratio | 0.5 | 0.5 |

Table 9: **Compute-performance trade-off analysis.** We compare the computational cost against the relative performance improvement over naive sampling. Results are reported for JavisDiT and MMDisCo using multi-verifier guidance. *NFE* denotes the Number of Function Evaluations.

| Model | Methods | Aggregation | # Samples | Computation | | Improvement (%) | | |
|---|---|---|---|---|---|---|---|---|
| | | | | Wall-clock (s) | NFE | Text ↑ | AV ↑ | Overall ↑ |
| JavisDiT | Naive Sampling | – | 1 | 49.76 | 30 | – | – | – |
| | Best-of-N | Weighted Sum | 5 | 258.06 | 150 | 23.75 | 22.76 | 23.33 |
| | | ARW | 5 | 258.21 | 150 | 20.69 | 27.98 | 23.81 |
| | EvoSearch | Weighted Sum | 5 | 691.19 | 400 | 27.93 | 27.82 | 27.88 |
| | | ARW | 5 | 691.59 | 400 | 22.93 | 36.31 | 28.67 |
| MMDisCo | Naive Sampling | – | 1 | 17.17 | 50 | – | – | – |
| | Best-of-N | Weighted Sum | 10 | 123.12 | 500 | 21.98 | 35.31 | 27.69 |
| | | ARW | 10 | 123.19 | 500 | 21.08 | 44.00 | 30.90 |
| | EvoSearch | Weighted Sum | 10 | 317.82 | 1300 | 31.67 | 58.62 | 43.22 |
| | | ARW | 10 | 318.06 | 1300 | 29.93 | 68.26 | 46.36 |

## B.3 Computational Overhead of Verifiers.

To provide a more practical view of multi-verifier ITS, we report the computational overhead of each verifier in Table 10. For text consistency, VideoReward is built on Qwen2-VL-2B (Wang et al., 2024b), whereas VQAScore uses the much larger CLIP-FlanT5-XXL (Chung et al., 2024) model. In addition, VQAScore

scores videos by evaluating individual frames and then averaging the resulting image-level scores, which further increases both inference time and TFLOPs compared to VideoReward. As a result, VQAScore is substantially more expensive than VideoReward. Nevertheless, as shown in Table 7, using VideoReward yields stronger text-consistency performance than VQAScore, indicating that the computationally efficient verifier can also be more effective in our setting. For audio-visual evaluation, both JavisScore and AVHScore use ImageBind-Huge (Girdhar et al., 2023) as the base model, showing similar computational overheads. However, as explained in Sec. 2.3, JavisScore shows better performance than AVHScore due to its evaluation at finer temporal granularity.

Table 10: **Computational overhead of individual verifiers.** We report the base model, number of parameters, inference time, and theoretical compute cost (TFLOPs) for each verifier.

| Category | Verifier | Base model | # Parameters | Inference time (s) | TFLOPs |
|---|---|---|---|---|---|
| Text-consistency | VideoReward | Qwen2-VL-2B | 2B | 0.13 | 4.81 |
|  | VQAScore | CLIP-FlanT5-XXL | 11B | 13.73 | 857.46 |
| AV-consistency | JavisScore | ImageBind-Huge | 1.2B | 1.26 | 33.69 |
|  | AVHScore | ImageBind-Huge | 1.2B | 1.22 | 33.10 |

## C  Adaptive Reward Weighting

### C.1  Algorithm Overview

Algorithm 1 outlines the complete procedure of our Adaptive Reward Weighting (ARW). The process operates in three distinct phases. First, in the *Reward Evaluation* phase, we compute raw scores for all generated candidates $\mathcal{X}$ using $K$ verifiers and update the historical buffer $\mathcal{H}_k$ to accumulate reward statistics. Second, during *Test-Time Calibration*, we optimize the learnable log-variance parameters $\mathcal{S}$ by minimizing the adaptive reward loss defined in Eq. 5. This step adapts the calibration scales $\sigma_k$ to the actual variance of the verifiers for the current input. Finally, in the *Aggregation and Selection* phase, the raw rewards are normalized by the learned scales and aggregated into a final score $R^{(i)}$. The candidate with the highest calibrated score is selected as the final output.

### C.2  Implementation of Adaptive Reward Weighting in EvoSearch

In evolutionary search, comparing candidates across different generation steps is challenging because the reward statistics are non-stationary; as new samples are accumulated, the underlying distribution shifts. Consequently, normalization factors derived from early generations may be incompatible with those of later stages, making direct comparison inconsistent.

To ensure fair comparison across the entire search trajectory, we implement ARW with a *re-scoring* strategy. We maintain a cumulative history buffer $\mathcal{H}_k$ that stores all raw reward observations for each verifier throughout the search process. At the end of each generation, we perform two critical updates:

1. **Global Adaptation:** We optimize the learnable log-variance parameters $\mathcal{S}$ by minimizing the adaptive reward loss (Eq. 5) over the entire cumulative history. This adapts the calibration scales to the global variance of all collected samples, rather than relying on the local statistics of a specific generation step.

2. **Unified Re-scoring:** We subsequently re-evaluate the weighted scores of the entire search trajectory for the current prompt using the newly updated parameters.

This ensures that the elite selection mechanism considers the full search history under a consistent weighting scheme, preventing outliers from being unfairly preserved due to provisional statistics from earlier stages.

---

**Algorithm 1** Adaptive Reward Weighting

---

**Input:** Set of candidates $\mathcal{X} = \{x^{(1)}, \ldots, x^{(N)}\}$
  Set of $K$ verifiers $\{V_k\}_{k=1}^{K}$ and historical buffer $\mathcal{H}_k$
  Learnable log-scale parameters $\mathcal{S} = \{s_1, \ldots, s_K\}$,
  Preference weights $\{w_k\}$, learning rate $\alpha$, iterations $T$, constant $\epsilon$

  ***Phase 1: Reward Evaluation and Collection***
 1: **for** each candidate $x^{(i)} \in \mathcal{X}$ **do**
 2:   **for** each verifier $k \in \{1, \ldots, K\}$ **do**
 3:     $r_k^{(i)} \leftarrow V_k(x^{(i)})$                          ▷ Compute raw reward
 4:     $\mathcal{H}_k \leftarrow \mathcal{H}_k \cup \{r_k^{(i)}\}$          ▷ Update history buffer
 5:   **end for**
 6: **end for**

  ***Phase 2: Test-Time Calibration***
 7: **for** step $t = 1$ to $T$ **do**                          ▷ Lightweight optimization loop
 8:   $\mathcal{L} \leftarrow 0$
 9:   **for** each verifier $k \in \{1, \ldots, K\}$ **do**
10:     Calculate $\widehat{\mathrm{Var}}(r_k)$ using $\mathcal{H}_k$                          ▷ Eq. 4
11:     $\mathcal{L} \leftarrow \mathcal{L} + \frac{1}{2}\exp(-s_k)\widehat{\mathrm{Var}}(r_k) + \frac{1}{2}|s_k|$          ▷ Adaptive Reward Loss (Eq. 5)
12:   **end for**
13:   $\mathcal{S} \leftarrow \mathcal{S} - \alpha\nabla_{\mathcal{S}}\mathcal{L}$                          ▷ Update log-scale parameters
14: **end for**

  ***Phase 3: Reward Aggregation and Selection***
15: **for** each candidate $x^{(i)} \in \mathcal{X}$ **do**
16:   $R^{(i)} \leftarrow 0$
17:   **for** each verifier $k \in \{1, \ldots, K\}$ **do**
18:     $\sigma_k \leftarrow \sqrt{\exp(s_k)}$                          ▷ Recover scale
19:     $R^{(i)} \leftarrow R^{(i)} + w_k \cdot \frac{r_k^{(i)}}{\sigma_k + \epsilon}$          ▷ Weighted sum of calibrated rewards (Eq. 3)
20:   **end for**
21: **end for**
22: **return** $x^{(i^*)}$ where $i^* = \arg\max_i R^{(i)}$

---

# D Ablation Study

## D.1 Ablation on Audio-Video Synchronization Verifiers

Table 11 compares two synchronization verifiers used as optimization targets under the same Best-of-$N$ pipeline on MMDisCo: AV-align (Yariv et al., 2024) versus JavisScore (JS), both paired with the text–video consistency verifier (VR). The results reveal a clear contrast in how the two verifiers guide the search. Optimizing VR + AV-align yields the largest gain on the AV-align metric itself. However, this improvement does not translate into broader quality gains: both text-related and AV-related metrics improve only marginally compared to naive sampling.

In contrast, optimizing VR + JS produces substantially stronger and more balanced improvements across metrics. While JS does not explicitly optimize AV-align, it leads to large gains in both AV-consistency and AV-sync, and it also preserves strong text-consistency improvements. This suggests that JS provides a more informative and transferable synchronization signal for guiding multimodal generation, whereas AV-align appears to be narrower and less aligned with the holistic objectives captured by our evaluation suite.

Table 11: **Ablation on audio-video synchronization verifiers.** We compare the effectiveness of JavisScore and AV-align as optimization targets under the Best-of-N algorithm on MMDisCo. AV-align is a metric used to evaluate audio-visual synchronization quality.

| Setup | Text-Consistency | | | | AV-Consistency | | AV-Sync | | Improvement (%) | | |
|---|---|---|---|---|---|---|---|---|---|---|---|
| | VR ↑ | VQA ↑ | TV-IB ↑ | TA-IB ↑ | AV-IB ↑ | AVH ↑ | Javis ↑ | AV-align ↑ | Text ↑ | AV ↑ | Overall ↑ |
| Naive Sampling | −1.104 | 0.597 | 0.300 | 0.179 | 0.189 | 0.182 | 0.160 | 0.547 | – | – | – |
| VR + AV-align | −0.672 | 0.638 | 0.310 | 0.177 | 0.196 | 0.187 | 0.166 | 0.733 | 12.05 | 3.40 | 8.34 |
| VR + JS | −0.591 | 0.653 | 0.315 | 0.221 | 0.264 | 0.259 | 0.240 | 0.559 | **21.08** | **44.00** | **30.90** |

## D.2 Sensitivity Analysis of Preference Weights

To validate the controllability of our proposed ARW, we conduct a sensitivity analysis on the preference weights $w_k$ defined in Eq. 3. As shown in Fig. 7, we observe a trade-off between text-consistency and AV-related improvements as we vary $w_{JS}$ from 0.3 to 0.7. As expected, increasing the weight for JS leads to a substantial improvement in AV-related metrics (orange line), rising from approximately 53% to 76%. This confirms that $w_k$ effectively acts as a steering knob, allowing the optimization process to prioritize the synchronization signal when desired. Conversely, the text improvement (blue line) exhibits a slight decline as the focus shifts toward AV objectives. However, this trade-off is remarkably stable; the text performance decreases gracefully rather than collapsing, maintaining a solid improvement of over 25% even when the weight assigned to JS is maximized. This indicates that ARW is robust to preference-weight changes and mitigates the *dominating reward* issue often observed in naive summation. Overall, ARW not only balances heterogeneous rewards automatically via calibration but also offers flexible user control over generation objectives through explicit preference weights.

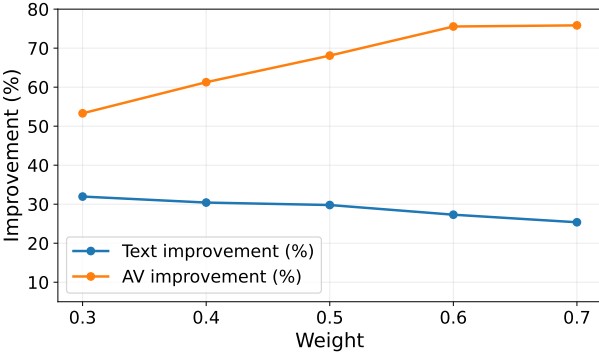

Figure 7: **Sensitivity analysis of preference weights.** We vary the preference weights in Eq. 3 from 0.3 to 0.7 in increments of 0.1. Increasing the weight for VideoReward-TA improves text consistency, whereas emphasizing JavisScore enhances AV consistency. Notably, the method demonstrates robustness, maintaining stable performance in both modalities without sudden degradation across the tested range.

## D.3 Hyperparameter Tuning for Weighted Sum Aggregation.

Table 12 presents the tuning results for the Weighted Sum baseline. We swept the balancing weight between text and audio-visual rewards from 0.2 to 0.8 to identify the optimal configuration. The results indicate that a weight of 0.2 yields the highest overall improvement for both Best-of-N and EvoSearch. Based on this empirical evidence, we adopt a weight of 0.2 for all Weighted Sum experiments reported in the main paper to ensure a strong and fair comparison.

Table 12: **Hyperparameter tuning for Weighted Sum aggregation.** We evaluate the performance by varying the balancing weight between VideoReward-TA and JavisScore in increments of 0.2. The configuration yielding the highest overall improvement is selected for the experiments in the main paper.

| Methods | Weight | Text-Consistency | | | | AV-Consistency | | AV-Sync | Improvement (%) | | |
|---|---|---|---|---|---|---|---|---|---|---|---|
| | | VR ↑ | VQA ↑ | TV-IB ↑ | TA-IB ↑ | AV-IB ↑ | AVH-Score ↑ | JavisScore ↑ | Text ↑ | AV ↑ | Overall ↑ |
| Naive sampling | – | −1.104 | 0.597 | 0.300 | 0.179 | 0.189 | 0.182 | 0.160 | – | – | – |
| **Best-of-N** | | | | | | | | | | | |
| BON | 0.2 | −0.513 | 0.665 | 0.317 | 0.210 | 0.248 | 0.244 | 0.225 | 21.98 | 35.30 | **27.69** |
| BON | 0.4 | −0.439 | 0.671 | 0.320 | 0.195 | 0.226 | 0.217 | 0.199 | 22.06 | 21.06 | 21.63 |
| BON | 0.6 | −0.430 | 0.673 | 0.320 | 0.188 | 0.216 | 0.206 | 0.187 | 21.37 | 14.78 | 18.55 |
| BON | 0.8 | −0.422 | 0.677 | 0.320 | 0.185 | 0.210 | 0.200 | 0.181 | 21.30 | 11.38 | 17.05 |
| **EvoSearch** | | | | | | | | | | | |
| EvoSearch | 0.2 | −0.312 | 0.681 | 0.322 | 0.239 | 0.288 | 0.285 | 0.267 | 31.67 | 58.62 | **43.22** |
| EvoSearch | 0.4 | −0.235 | 0.689 | 0.324 | 0.212 | 0.259 | 0.251 | 0.233 | 30.14 | 40.19 | 34.45 |
| EvoSearch | 0.6 | −0.205 | 0.686 | 0.324 | 0.191 | 0.222 | 0.215 | 0.197 | 27.76 | 19.57 | 24.25 |
| EvoSearch | 0.8 | −0.210 | 0.688 | 0.325 | 0.182 | 0.205 | 0.197 | 0.180 | 26.56 | 9.74 | 19.35 |

## D.4 Convergence behavior of ARW

To improve the interpretability of ARW, we additionally analyze its convergence behavior under different optimizers in Fig. 8. We compare Adam (Kingma & Ba, 2014), SGD (Robbins & Monro, 1951), and RMSprop (Tieleman, 2012) by tracking both the calibration loss ($L_{arw}$) and the learned scale parameters over optimization steps. All three optimizers converge to very similar final loss values and nearly identical scale parameters, indicating that ARW is insensitive to the choice of optimizer. In particular, the learned scales stabilize within roughly 50–100 steps in all cases, suggesting that ARW is a straightforward and generalizable algorithm that is easy to implement. Among the tested optimizers, Adam exhibits the smoothest and most stable convergence, while SGD and RMSprop also converge reliably after a rapid initial decrease. Based on this observation, we use Adam as the default optimizer in the main experiments due to its stable optimization dynamics, while noting that the final behavior of ARW is robust across optimizer choices.

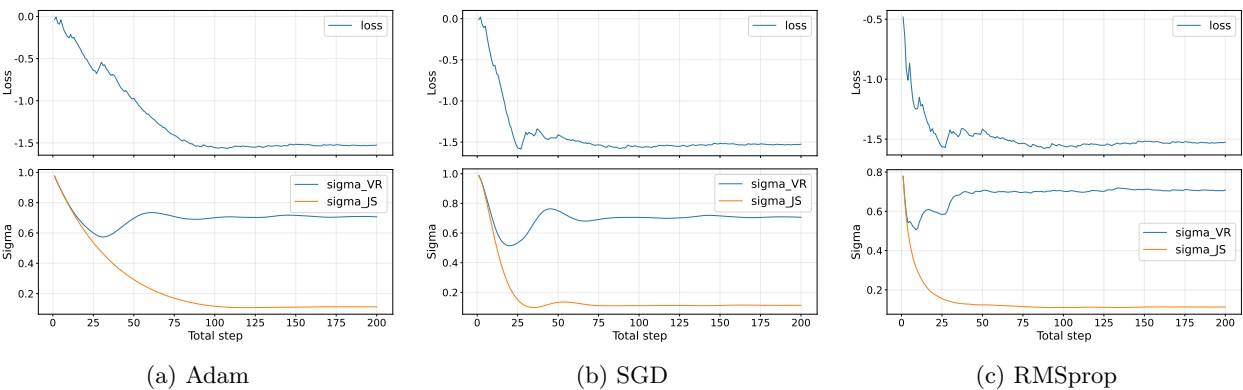

| (a) Adam | (b) SGD | (c) RMSprop |
|---|---|---|

Figure 8: **Convergence behavior of ARW under different optimizers.** We plot the calibration loss (top) and the learned scale parameters (bottom) for Adam, SGD, and RMSprop. All optimizers converge to similar final solutions, although their early optimization dynamics differ. Overall, the results indicate that ARW is stable and largely insensitive to optimizer choice.

## D.5 Generalization to a Stronger AV-Generation Model

To examine whether our framework generalizes beyond the backbones used in the main paper, we additionally evaluate it on LTX-2 (HaCohen et al., 2026), a strong open-source joint audio-video generation model. We

Table 13: **Performance of LTX-2 using ITS.** Text improvement and AV improvement are defined as the average performance gains over naive sampling for text-related and AV-related metrics, respectively.

| Aggregation | Text-Consistency | | | | AV-Consistency | | AV-Sync | Improvement (%) | | |
|---|---|---|---|---|---|---|---|---|---|---|
| | VR* ↑ | VQA ↑ | TV-IB ↑ | TA-IB ↑ | AV-IB ↑ | AVH-Score ↑ | JavisScore* ↑ | Text ↑ | AV ↑ | Overall ↑ |
| Naive sampling | 0.503 | 0.908 | 0.277 | 0.174 | 0.262 | 0.253 | 0.222 | – | – | – |
| ARW (Ours) | 0.639 | 0.915 | 0.279 | 0.183 | 0.274 | 0.267 | 0.238 | 8.41 | 5.76 | 7.28 |

conduct the experiment on the JavisBench-mini benchmark under the Best-of-$N$ setting. We compare naive sampling and multi-verifier ITS with ARW using the same evaluation protocol as in the main paper.

As shown in Table 13, ARW consistently improves over naive sampling on LTX-2, achieving +8.41% text improvement, +5.76% AV improvement, and +7.28% overall improvement. We further provide qualitative comparisons in Fig. 14 and Fig. 15, which show that ARW improves prompt-faithful audiovisual generation on LTX-2. In Fig. 14, ARW better satisfies fine-grained textual constraints: in the top example, it generates four birds with attributes that better match the prompt, whereas naive sampling generates only three birds and fails to reflect the red-head attribute; in the bottom example, ARW better follows the specified left-to-right helicopter motion, while naive sampling does not clearly satisfy the requested trajectory. In Fig. 15, ARW also improves audio-visual grounding: in the piano example, it better captures the intended playing posture and interaction with the instrument, and in the walking example, it generates more distinct footstep events that are temporally aligned with the motion of the two figures. These results suggest that the benefit of ARW is not limited to the original backbones used in the main paper. Instead, ARW also transfers to a stronger AV-generation model, supporting our claim that it is a backbone-agnostic inference-time framework rather than a model-specific technique. The video samples are available at the following link.

# E   Qualitative Results

In this section, we provide extended qualitative results for both JavisDiT and MMDisCo. First, for JavisDiT, we evaluate the effectiveness of our approach using complex prompts from JavisBench-mini, comparing it against naive sampling and single-verifier (VR) guidance. Additionally, for MMDisCo, we visualize the generation quality on the VGGSound test set.

**Qualitative Analysis on JavisDiT.** As shown in Fig. 9, the prompt imposes fine-grained constraints: a count of "two pigeons" and a detailed description of a "black mesh fence." Baseline methods struggle to satisfy these constraints simultaneously. Naive sampling fails to adhere to the numerical constraint, incorrectly generating three pigeons instead of two. Similarly, while the single-verifier approach generates the correct number of subjects, it fails to generate the requested "mesh fence." In contrast, ARW successfully satisfies all conditions, generating exactly two pigeons behind a correctly rendered black mesh fence.

In Fig. 10, the prompt describes a static visual scene alongside a specific audio event ("An owl hoots rhythmically"). Naive sampling fails to generate the requested audio event, producing a spectrogram dominated by constant noise rather than distinct calls. Notably, the single-verifier model exhibits a critical failure mode we term "modality leakage." Because VR optimizes solely for text-video alignment, it misinterprets the auditory description "An owl hoots" as a visual instruction, resulting in the hallucination of a "bird's wing" appearing in the window. It attempts to visually render the sound source rather than treating it as an auditory element. In contrast, ARW correctly disentangles these multimodal constraints. It preserves a clean visual scene while successfully generating the "rhythmic owl hoots" in the audio track, as evidenced by the distinct intermittent patterns in the spectrogram. This confirms that ARW effectively uses JS to enforce audio-specific constraints without corrupting the visual generation with unnecessary artifacts.

**Qualitative Analysis on MMDisCo.** As shown in Fig. 11, ITS significantly improves both physical plausibility and semantic alignment compared to naive sampling. In the top example ("Black-capped chickadee calling"), the naive method suffers from severe anatomical distortion, generating a bird with its head twisted backwards, whereas applying ITS rectifies this artifact to produce a photorealistic bird with correct anatomy.

Furthermore, the spectrogram confirms that ITS generates distinct "calling" patterns synchronized with the visual content, unlike the unclear baseline audio. Similarly, in the bottom example ("Playing congas") which requires depicting an action, naive sampling fails to capture the verb "playing" and generates only a static instrument; in contrast, ITS method correctly generates the agent performing the action by showing hands striking the drums. These results collectively demonstrate that our method effectively reinforces text-video alignment, ensuring that both objects and their associated actions are faithfully rendered.

**Robustness to Prompt Variability.** In real-world applications, user prompts are rarely as structured or grammatically pristine as those found in standard benchmarks. To evaluate the robustness of our ITS framework against prompt variability, we conducted a stress test using unstructured text inputs. Specifically, we utilized ChatGPT 4.0 to systematically perturb the original prompts from the JavisBench-mini dataset into three common real-world styles: a conversational style expanded with filler and verbose instructions (e.g., "Umm, can you generate a video where..."), a fragment style featuring highly compressed and disjointed keywords that omit conjunctions and articles, and a grammar typo style injected with realistic spelling and grammatical errors.

Qualitative results indicate that standard generation pipelines relying on naive sampling are sensitive to prompt structure, whereas our proposed ITS framework maintains robust text-video-audio alignment. Even on standard prompts, naive sampling often misses fine-grained details. For instance, as shown in Fig. 12, the baseline fails to generate specific textual elements such as a "pouring stream," "ripples," or "bubbles," whereas the ITS framework successfully identifies and selects candidates that faithfully reproduce these detailed semantic components. Fig. 13 further highlights how variations in prompt style exacerbate the failures of the baseline model. When presented with a fragment-style prompt, naive sampling suffers from attribute leakage, incorrectly generating an orange bird. Furthermore, a minor typographical error, such as spelling "forest" as "forst," causes the baseline to completely fail at generating the appropriate background environment. In contrast, our ITS framework consistently synthesizes high-quality audio-video pairs across all perturbation styles. By dynamically aggregating multiple verifier rewards, the proposed framework is robust to textual noise and structural anomalies, focusing on the core semantics to ensure that the generated output strictly adheres to the user's underlying intent.

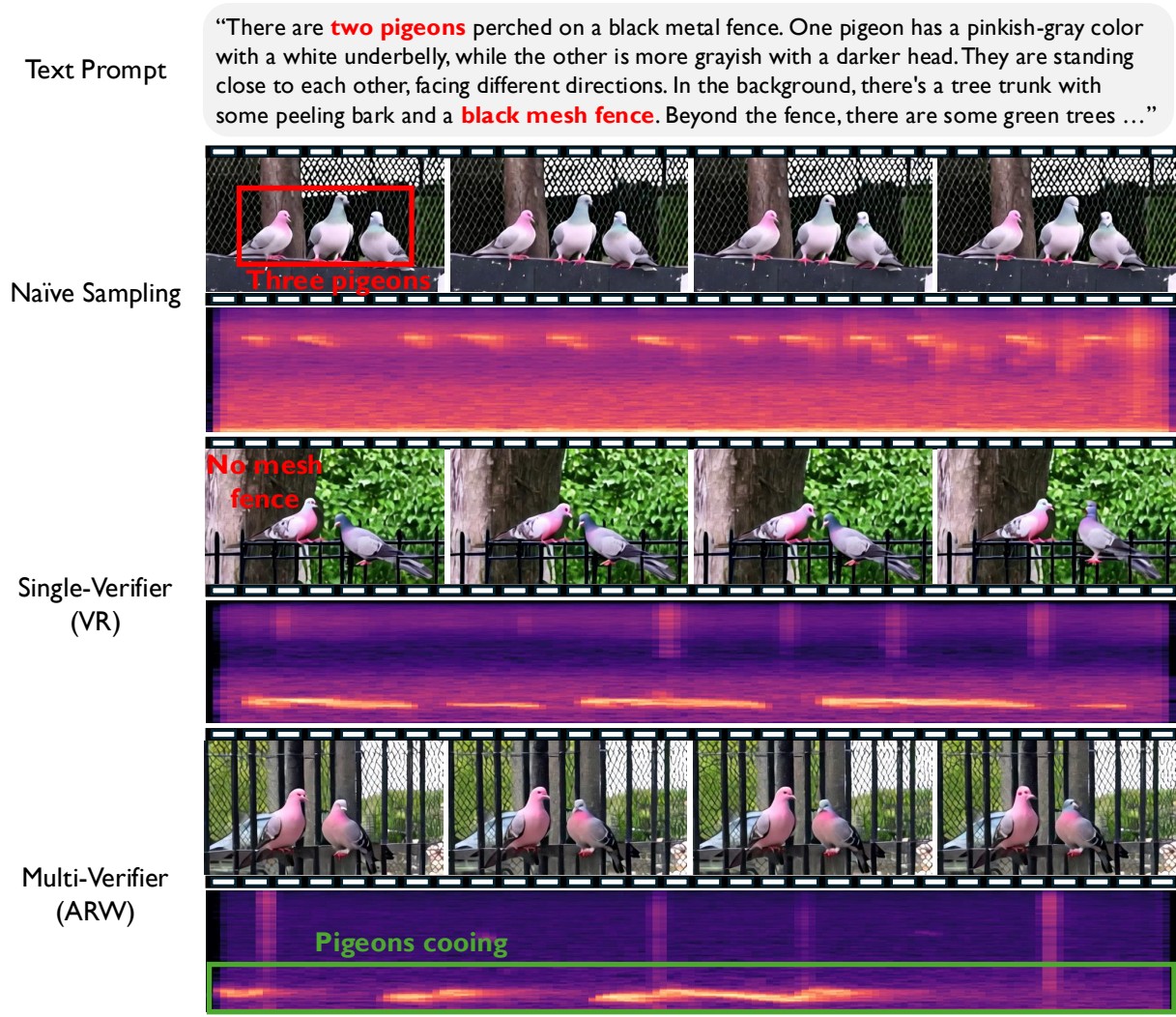

Figure 9: **Qualitative comparison of generated samples.** We compare the outputs of naive sampling, single-verifier (VR-guidance), and our multi-verifier (ARW) given a complex text prompt on JavisDiT. Naive sampling fails to respect the count constraint, generating three pigeons. Single-verifier guidance misses the fine-grained visual detail, failing to render the mesh fence. Multi-verifier successfully satisfies all constraints, accurately generating two pigeons with the correct background texture while maintaining audio-visual alignment. The video samples are available at the following link.

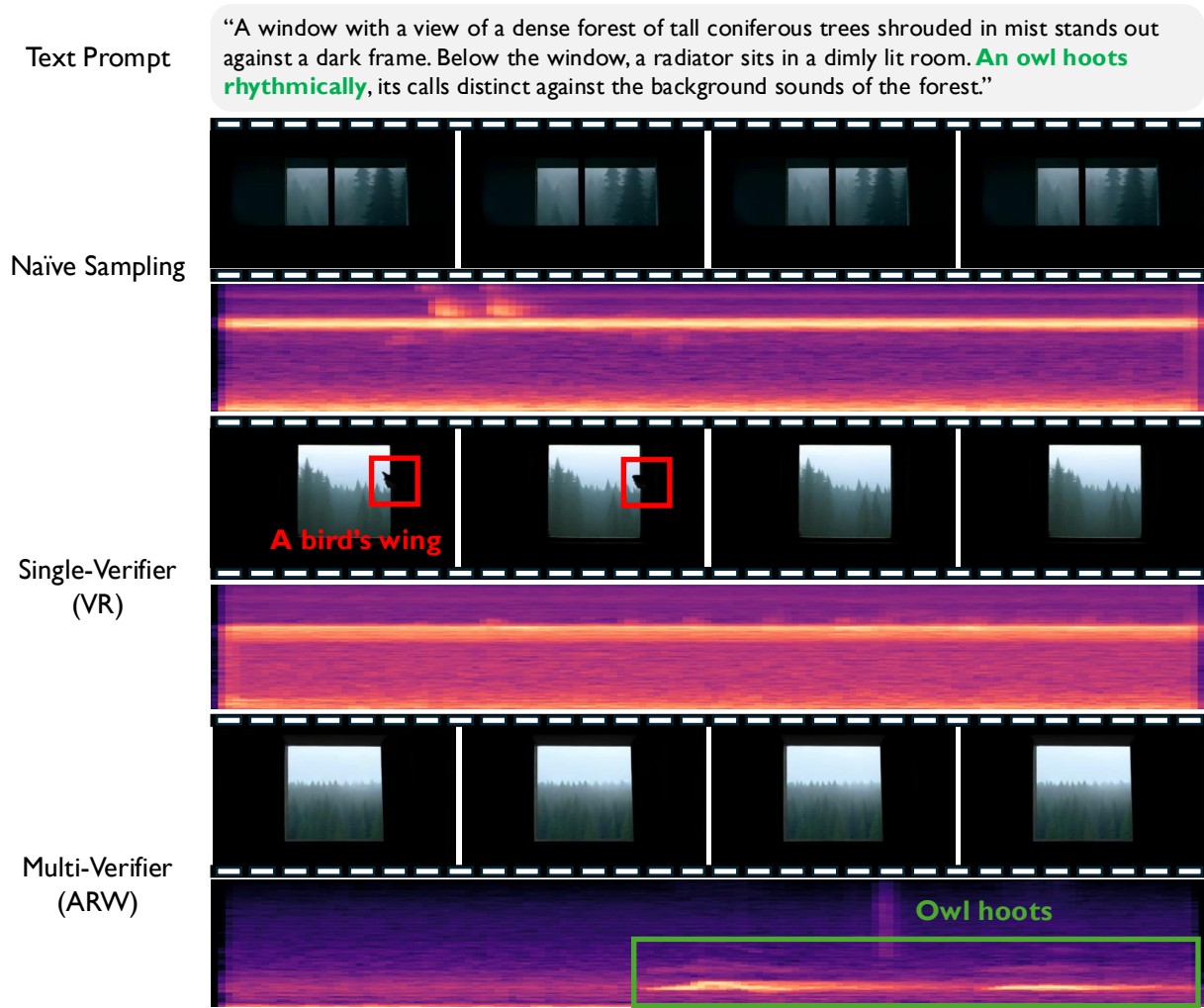

Figure 10: **Qualitative comparison of generated samples.** We compare the outputs of naive sampling, single-verifier (VR-guidance), and our multi-verifier (ARW) given a complex text prompt on JavisDiT. Naive sampling fails to generate the specific audio event. Single-verifier misinterprets the audio description as a visual cue, hallucinating a bird's wing in the video frame. Multi-verifier correctly assigns the semantic constraints to their respective modalities: it generates clear, rhythmic hooting sounds while keeping the visual scene consistent with the "A window with a view" description. The video samples are available at the following link.

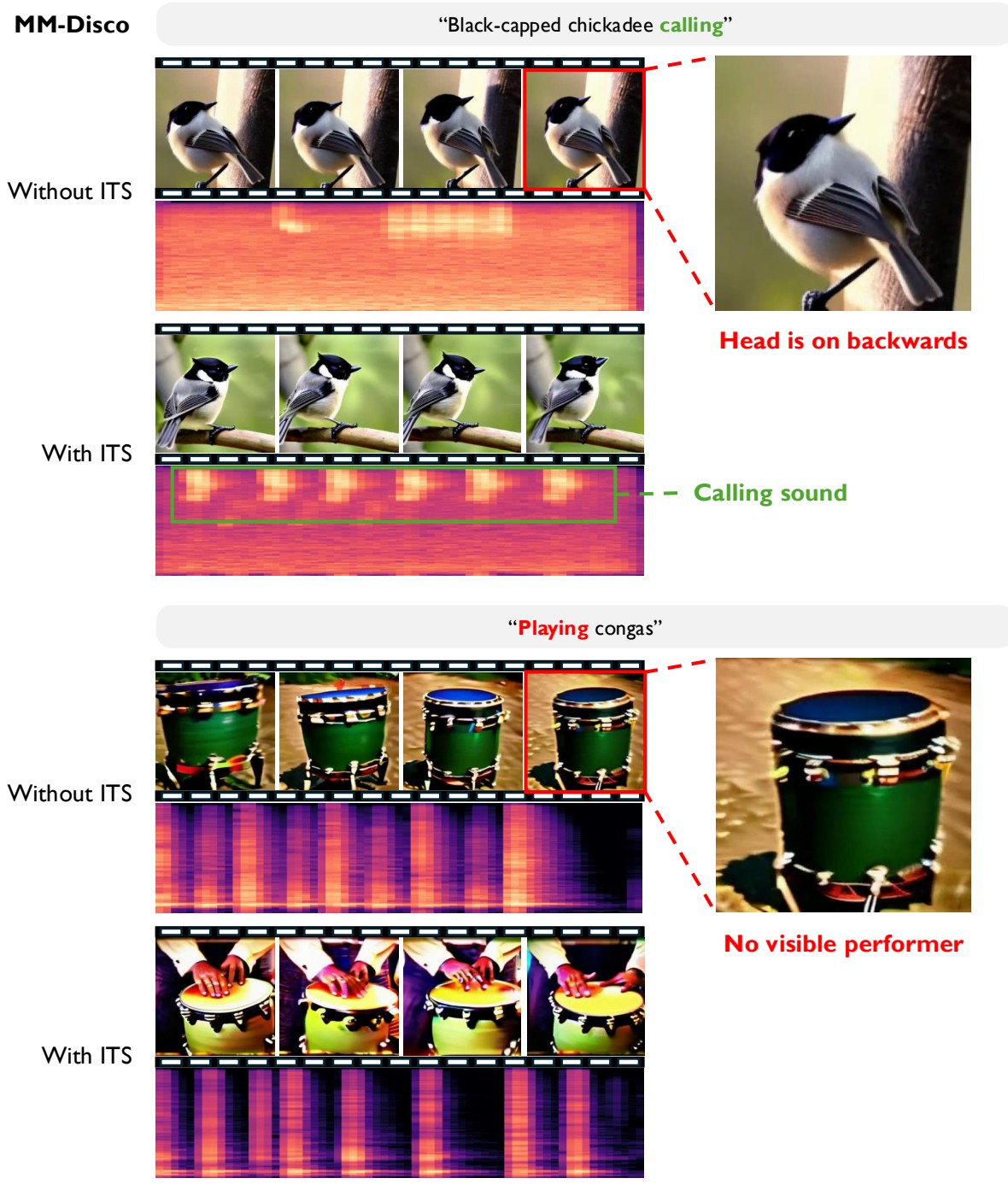

Figure 11: **Qualitative comparison of generated samples.** We compare the outputs of naive sampling and our multi-verifier (ARW) approach using text prompts from MMDisCo. The results demonstrate the correction of semantic and physical failures. In the top example, naive sampling generates a bird with severe anatomical distortion (head twisted backwards), whereas our method ensures physical correctness and generates synchronized calling sounds. In the bottom example, naive sampling fails to render the performer (missing performer), while our method correctly depicts the interaction by showing the player's hands on the instrument. The video samples are available at the following link.

| | |
|---|---|
| Original prompt | "Oil is being poured into a shiny stainless steel pot on a stove, creating ripples and small bubbles on the surface. The pot is reflective, and the stove has a digital control panel with orange buttons. As the oil hits the hot surface, there's a slight sizzling noise, accompanied by background music." |
| Conversational style | "I want a realistic video where oil is being poured into a shiny stainless steel pot on a stove, creating ripples and small bubbles on the surface. The pot is reflective, and the stove has a digital control panel with orange buttons. As the oil hits the hot surface, there's a slight sizzling noise, accompanied by background music." |
| Fragment style | "oil being poured into shiny stainless steel pot on stove, creating ripples, small bubbles on surface, pot reflective, stove digital control panel, orange buttons, oil hits hot surface, slight sizzling noise, accompanied by background music" |
| Grammar typo | "Oil being poured into a shiny stainless steel pot on a stove, creating ripples and smll bubbles on the surface. The pot is reflective, and the stove has a dgital control panel with orange buttons. As the oil hits the hot surface, there's a slight sizzling noise, accompanied by background music." |

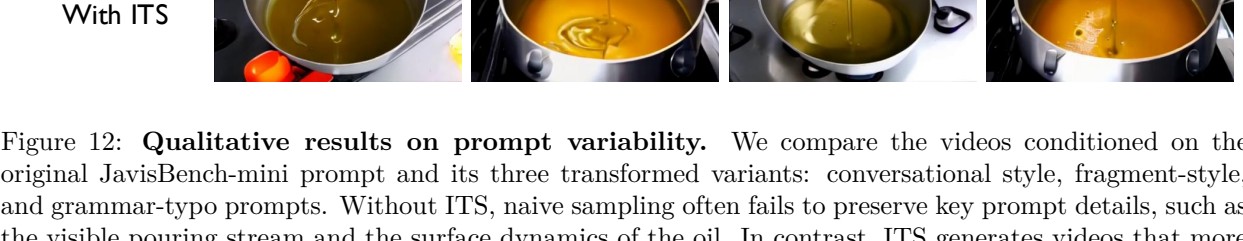

Figure 12: **Qualitative results on prompt variability.** We compare the videos conditioned on the original JavisBench-mini prompt and its three transformed variants: conversational style, fragment-style, and grammar-typo prompts. Without ITS, naive sampling often fails to preserve key prompt details, such as the visible pouring stream and the surface dynamics of the oil. In contrast, ITS generates videos that more consistently follow the intended semantics across all prompt variants, demonstrating improved robustness to realistic and unstructured prompt styles. The video samples are available at the following link.

| Original prompt | "Two colorful birds with vibrant blue, green, and white feathers are perched on a branch, facing each other and interacting. Their tails are long and feature black and white stripes. A lush green forest with blurred foliage serves as the background, and a bird chirping sound can be heard throughout. |
| Conversational style | "Umm, can you generate a super realistic video where two colorful birds with vibrant blue, green, and white feathers are perched on a branch, facing each other and interacting. Their tails are long and feature black and white stripes. A lush green forest with blurred foliage serves as the background, and a bird chirping sound can be heard throughout? Please make the audio match naturally too. |
| Fragment style | "two colorful birds, vibrant blue, green, white feathers perched on branch, facing each other, interacting, their tails long, feature black, white stripes, lush green forest, blurred foliage serves, background, bird chirping sound throughout" |
| Grammar typo | "Two colorful birds with vibrant blue, green, and white feathers are perched on a branch, facing each other and interacting. Thier tails are long and feature black and white stripes. A lush green forst with blurred foliage serves as the background, and a bird chirping sound can be heard thrughout." |

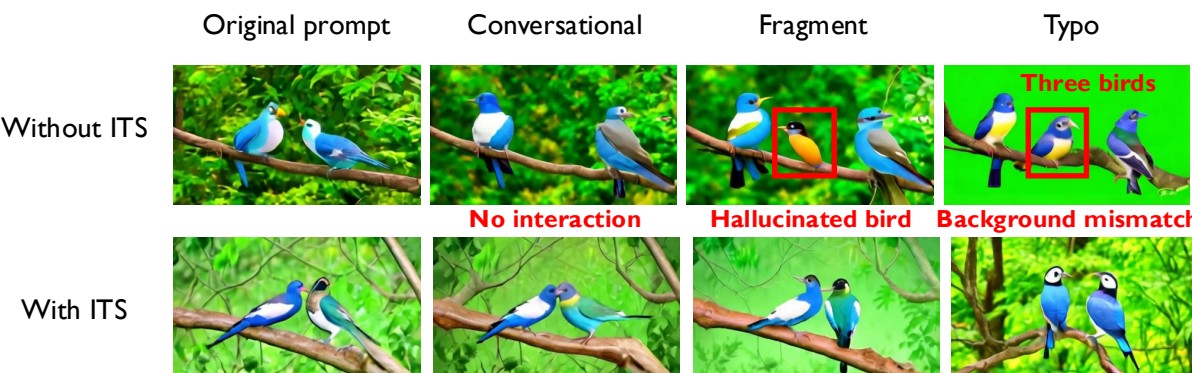

Figure 13: **Qualitative results on prompt variability.** We compare the outputs conditioned on the original JavisBench-mini prompt and its three transformed variants: conversational style, fragment-style, and grammar-typo prompts. Without ITS, naive sampling often fails to preserve key prompt semantics under prompt variation: it loses the intended interaction between the two birds, hallucinates an additional orange bird in the fragment-style case, and fails to generate the intended forest background under the typo-corrupted prompt. In contrast, ITS generates videos that more consistently preserve the core semantics across all prompt variants, including the two birds, their interaction, and the lush green forest background. The video samples are available at the following link.

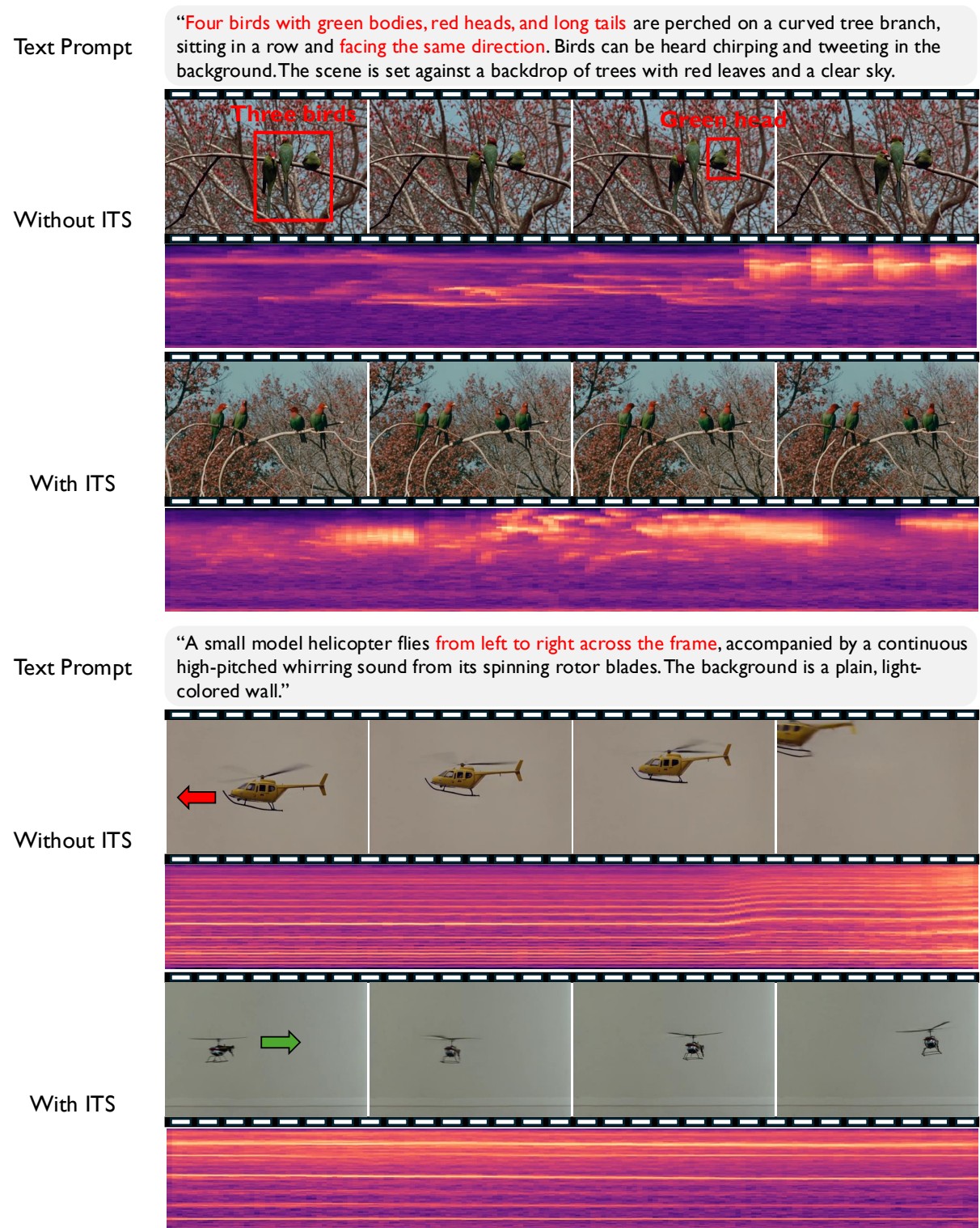

Figure 14: **Qualitative comparison on LTX-2.** We compare naive sampling and multi-verifier ITS with ARW on a stronger open-source joint audio-video generation model. **Top:** ITS better satisfies the fine-grained bird attributes and count. **Bottom:** ITS better follows the intended helicopter motion direction. These examples suggest that ARW improves prompt-faithful audiovisual generation on LTX-2 as well. The video samples are available at the following link.

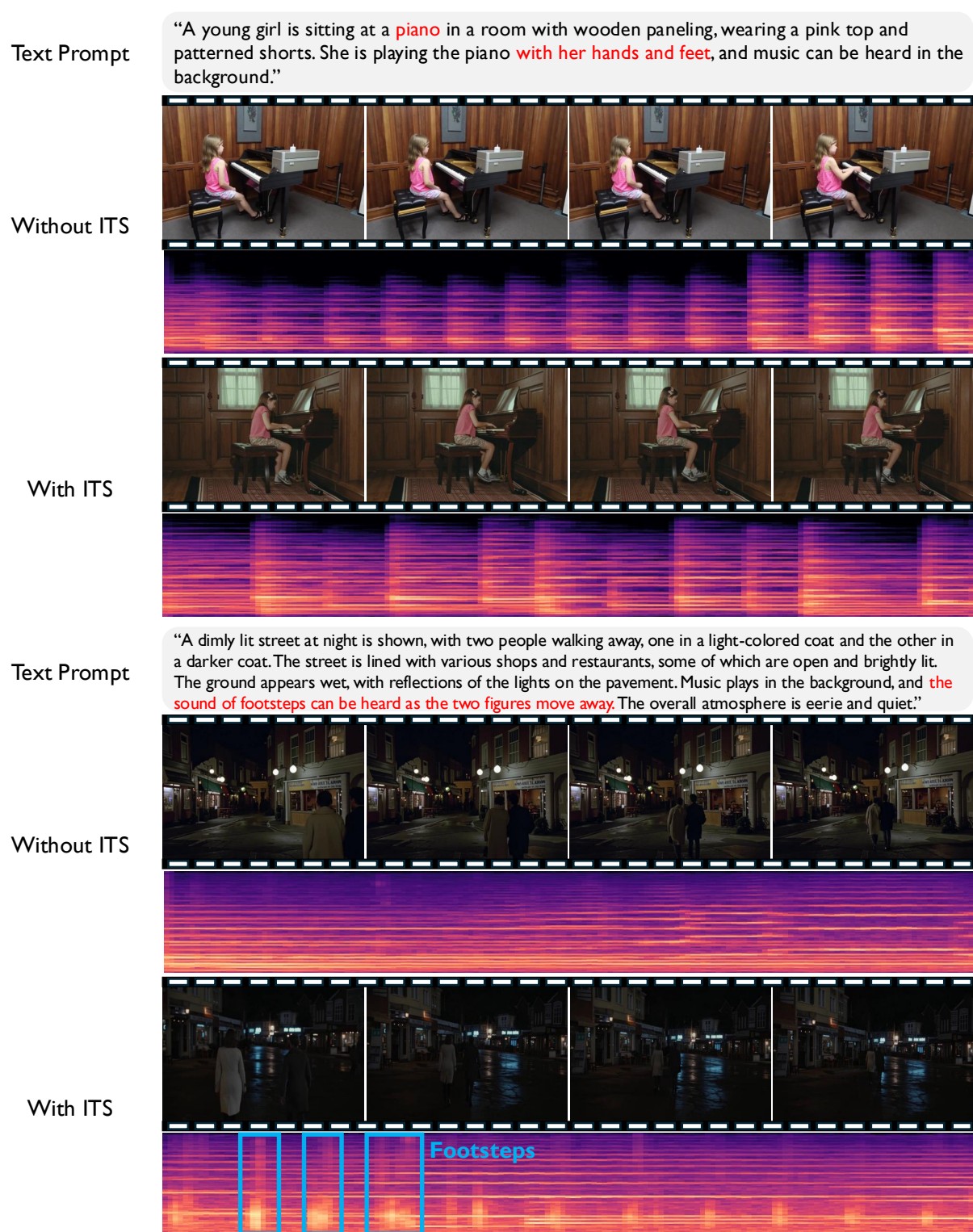

Figure 15: **Qualitative comparison on LTX-2.** We compare naive sampling and multi-verifier ITS with ARW on a stronger open-source joint audio-video generation model. **Top:** ITS better reflects the intended piano-playing posture and interaction with the instrument. **Bottom:** ITS produces more distinct footstep events that better align with the walking motion. These examples suggest that ARW improves prompt-faithful audiovisual generation on LTX-2 as well. The video samples are available at the following link.

## F  Asset Licenses

The licenses of the assets used in the experiments are denoted as follows:

**Models.**

- **JavisDiT** (Liu et al., 2026): `https://github.com/JavisVerse/JavisDiT`
- **MMDisCo** (Hayakawa et al., 2025): `https://github.com/SonyResearch/MMDisCo`
- **LTX-2** (HaCohen et al., 2026): `https://github.com/Lightricks/LTX-2`

**Implementation Resources.**

- **VQAScore** (Lin et al., 2024): `https://github.com/linzhiqiu/t2v_metrics`
- **JavisScore** (Liu et al., 2026): `https://huggingface.co/datasets/JavisVerse/JavisBench`
- **VideoReward** (Liu et al., 2025b): `https://github.com/KlingTeam/VideoAlign`
- **VBench** (Huang et al., 2024): `https://github.com/Vchitect/VBench`
- **AV-align** (Yariv et al., 2024): `https://github.com/guyyariv/TempoTokens`
- **EvoSearch** (He et al., 2025a): `https://github.com/tinnerhrhe/EvoSearch-codes`

## G  Broader Impact

This paper advances joint audio–video generation by demonstrating that inference-time scaling can substantially improve multimodal alignment and synchronization without additional training. By reducing reliance on costly retraining, the proposed framework can lower the barrier to deploying high-quality audio–video generation systems, potentially benefiting applications in content creation and immersive media such as AR/VR. In particular, this training-free quality improvement may help smaller organizations or researchers with limited computational resources to leverage powerful generative models more effectively. However, as realistic audio–video generation becomes increasingly accessible, it also highlights the need for robust detection and provenance mechanisms to mitigate potential misuse, including misinformation and deepfakes.

## H  Failure Cases

Although ITS consistently enhances the quality and text fidelity of generated outputs over naive sampling, it does not fully overcome the intrinsic limitations of current joint audio-video generation models. We present two representative failure modes in Fig. 16. First, **physical plausibility is still not guaranteed**. In the upper example, ITS improves the overall prompt alignment by generating a hummingbird feeding from the feeder, but the bird's wings still unrealistically pass through the feeder, indicating a violation of basic physics. Second, **unstable temporal consistency remains an issue**. In the lower example, naive sampling fails to follow the prompt faithfully, showing the horse moving leftward rather than trotting forward around the arena. ITS improves the naturalness of motion direction and generates a more prompt-faithful video, but flickering artifacts still remain across frames.

These examples suggest that the effectiveness of ITS is ultimately bounded by the base capability of the pretrained generator. In addition, the current verifiers mainly focus on semantic alignment, audio-visual consistency, and synchronization, but do not explicitly capture physical realism or temporal coherence. As a result, ITS can select samples that better satisfy the text prompt while still exhibiting implausible object interactions or frame-to-frame flickering. We believe that addressing these limitations will require not only stronger base models, but also verifiers that are more aware of physics and temporal consistency.

# I   Future Work

As discussed in our limitations, the potential of ITS for joint audio-video generation is currently constrained by high memory footprints and substantial computational overhead. To make scalable ITS practical for real-world applications, addressing the computational cost of generating and evaluating multiple candidates is particularly crucial.

To this end, developing efficient search strategies considering the characteristics of joint audio-video generation presents a highly promising direction. Existing literatures (Zhao et al., 2026; Oshima et al., 2025) suggest that, as denoising progresses, partially denoised intermediate states become increasingly informative of final video quality and semantic alignment, enabling intermediate rewards to guide search before full denoising. While our preliminary experiments confirmed this trend for visual semantics, we additionally discovered a unique challenge for joint audio-video generation: fine-grained multimodal alignment, such as audio-visual synchronization and text-audio consistency, only emerges in the later stages of the diffusion process and requires full denoising for accurate measurement.

Based on this finding, we propose that an efficient search strategy would be early pruning guided by a multimodal proxy verifier. If future research can develop a lightweight proxy model capable of predicting the final audio-visual synchronization from early-stage noisy latents, the framework could prune unpromising noise trajectories in the initial steps. This would drastically reduce the computational resources used for fully generating and evaluating low-quality candidates. We believe this direction provides valuable insights for practitioners and establishes a clear direction for optimizing multimodal ITS at scale.

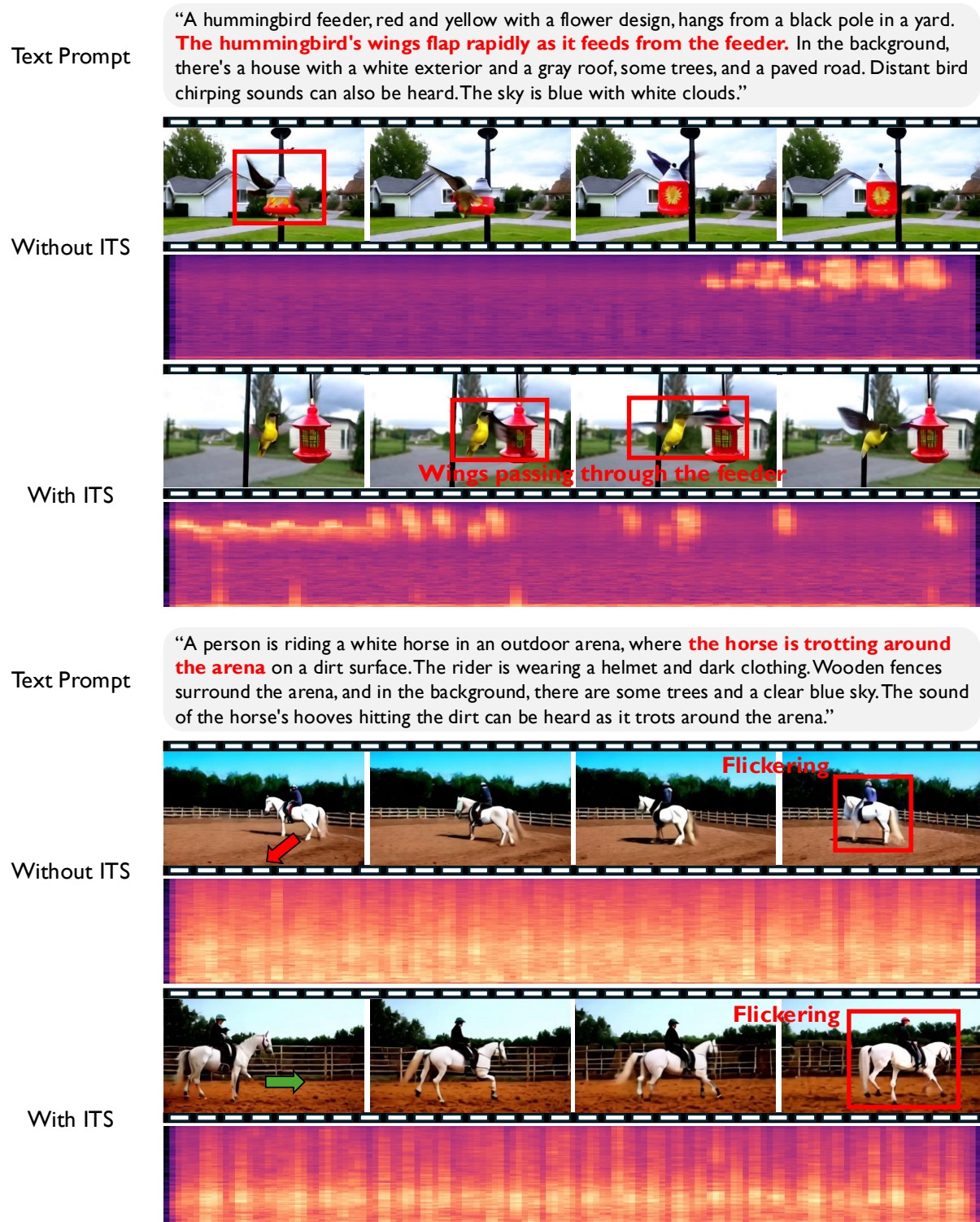

Figure 16: **Failure cases. Top:** Although the ITS framework generally improves the visual quality and text alignment, it still struggles with physical plausibility. For instance, the generated bird's wing unnaturally passes through the solid feeder, indicating a lack of physical priors. **Bottom:** ITS successfully corrects semantic errors, such as guiding the horse to properly walk forward, unlike the naive sampling which generates it walking in the wrong direction. However, temporal artifacts, such as flickering, still persist in the final output. The video samples are available at the following link.

