# OpenReview forum: "Inference-Time Scaling for Joint Audio-Video Generation"
_TMLR — Accepted by TMLR_

### Review · Reviewer_vXxV · 2026-02-28

**Summary Of Contributions:**

The authors find that current inference-time scaling (ITS) approaches are mostly limited in single-modality domains. In this view, the authors extend ITS from a single modality to multimodal domains for joint audio-video generation.

**Audience:**

Yes

**Audience Explanation:**

The area for Joint Audio–Video Generation still needs further development, and the introduction of inference-time scaling can somehow strengthen the performance. While the improvements are not extremely great, they can be considered as a promising future for additional training/steering during inference.

**Claims And Evidence:**

No

**Claims Explanation:**

1. The discussions on the four-verifier setting need further clarification. As shown in the paper, "adding multiple verifiers implicitly overweights audio-related objectives, leading to a bias that prioritizes audio-video alignment at the expense of prompt-faithful video generation." This observation is similar to the observation of using a single verifier, which "fails to achieve a balanced improvement across all metrics." Due to these similar explanations, a natural question will be: is this a balancing issue (same as a single verifier)? The current explanation is purely experimental-based, without further exploration.

2. The concept of ITS should be discussed carefully with the concept of test-time learning/training [1-4], which is a well-developed area for inference-time adaptation. Also, the authors should at least discuss the possibility of introducing reinforcement learning during inference, as supported by multiple recent studies [1,3,5,6].

3. The overall contribution of this paper is still quite limited. The ARW is fully supported by experimental evidence, while further novelties should be explored to strengthen the point (with the potential drawback of latency during test time). The authors need to convince me why I need to pay for the multi-verifier setting (tables 4 and 5 improvements are not sufficiently strong), with additional training on the learnable calibration parameter.

[1] Ttrl: Test-time reinforcement learning

[2] Efficient test-time adaptation of vision-language models

[3] On-the-Fly VLA Adaptation via Test-Time Reinforcement Learning

[4] Bafta: Backprop-free test-time adaptation for zero-shot vision-language models

[5] The entropy mechanism of reinforcement learning for reasoning language models

[6] Reinforcement learning for reasoning in large language models with one training example

**Requested Changes:**

Please see my above concerns. While the research draws upon experimental evidence, more support is needed to strengthen its claim. Also, the intro of ARW remains quite incremental.

---

> ### Author Response · Authors · 2026-03-11
> **Response to Reviewer vXxV - Part I**
>
> We thank the reviewer for the valuable time and effort to evaluate our paper. We hope our detailed responses below can help address the reviewer's concerns.
>
> > Q1. The discussions on the four-verifier setting need further clarification. Due to similar explanations, a natural question will be arised: is this a balancing issue (same as a single verifier)?
>
> **Ans:** We would like to clarify that the balancing issue in the single-verifier case is fundamentally different from that in the four-verifier case.
>
> The balancing issue in the single-verifier setting is characterized by a failure to improve non-targeted metrics. For instance, utilizing semantic guidance alone yields only marginal or limited gains for audio-video alignment, and vice versa. In other words, the reward signal is restricted to a single objective, so improvements remain confined to that target metric while the other modality remains unguided and vulnerable to verifier hacking.
>
> In contrast, the four-verifier setting does achieve performance improvements across all metrics compared to the naive sampling baseline and better overall performance gain compared to the single verifier. The "skew" we referred to does not imply a complete failure to improve text-consistency. Rather, it indicates that the magnitude of improvement in audio-video metrics is disproportionately larger than the improvement in text-consistency.
> This specific imbalance is driven by the semantic redundancy among the evaluation rewards. Specifically, the reward signals are not fully independent, as several verifiers provide partially overlapping supervision during candidate selection. For instance, both JavisScore and AVHScore utilize ImageBind to extract audio and visual embeddings, relying on these shared representations to compute their reward functions. Therefore, the issue here is not a complete neglect of non-targeted metrics as in the single verifier.
>
> > Q2. The relationship between ITS and prior work on test-time adaptation/training is not clearly discussed, and the paper should also comment on whether reinforcement learning during inference could be a relevant extension.
>
> **Ans:** We sincerely thank the reviewer for providing these insightful references. We clarify that the scope of our proposed method in the context of ITS is different from test-time adaptation and test-time RL.
>
> In our setting, the generator and verifiers remain fixed, and inference-time compute focuses on search-and-selection over candidates. ARW does not adapt the generator/policy to test data; it only updates lightweight reward-calibration parameters for adaptive normalization across heterogeneous verifiers and candidate ranking. In contrast, prior test-time adaptation methods adapt the predictor to new data distributions, either through parameter updates or auxiliary adaptive states (e.g., caches or centroids), while test-time RL explicitly refines the policy/model using reward feedback. RL-based inference-time adaptation is an interesting future direction, but it is distinct from the training-free ITS setting studied in this work. Our main contribution is to introduce a straightforward yet effective ITS framework tailored to T2AV generation, where the key challenge is to improve generation quality by balancing multiple heterogeneous verifier signals without modifying the underlying generator. We will elaborate on the comparison of ITS with TTA and test-time RL in the related work section.

---

> ### Author Response · Authors · 2026-03-11
> **Response to Reviewer vXxV - Part  II**
>
> > Q3. The paper’s overall novelty remains limited, and the practical value of ARW is not yet fully convincing given the seemingly modest gains in Tables 4–5 and the potential concern about additional test-time overhead from the learnable calibration parameters.
>
> **Ans:** First of all, we want to emphasize the contributions of our paper as follows:
>
> 1. We conduct the first comprehensive study of ITS for joint audio–video generation, revealing the limitations of single-verifier guidance and demonstrating the need for a multi-verifier framework to achieve balanced improvements in overall generation quality.
> 2. By systematically exploring different reward combinations, we identify the most effective multi-verifier configuration that balances heterogeneous objectives.
> 3. We introduce Adaptive Reward Weighting (ARW), a novel test-time optimization method for aggregating heterogeneous reward signals. Our approach enables stable multi-objective selection without requiring prior knowledge of reward distributions or any offline calibration.
>
> We further clarify the latency and effectiveness of ARW as follows:
>
> - ARW does not involve training a neural network with a large number of parameters. It merely updates a few scalar calibration parameters using a lightweight optimizer. As shown in Table 9, the wall-clock time difference between ARW and the naive Weighted Sum is an increase of only 0.06% to 0.08%. The overhead is negligible relative to the diffusion sampling process.
> - In addition, Table 12 shows that the Weighted Sum baseline requires manual tuning to find the optimal weight for each reward signal. While Z-score normalization is competitive with ARW, it relies on pre-computed reward statistics from a training set, making it vulnerable to distribution shifts and less practical when such statistics are not available at test time. In contrast, ARW achieves comparable or better performance without requiring any prior statistics. Moreover, compared with other statistic-free baselines such as Rank, Min-Max, and Weighted Sum, ARW consistently yields stronger performance improvements across benchmarks.
>
> Finally, the core reason for paying this minimal cost is to address “*verifier hacking”*, a critical vulnerability of single-verifier guidance. Our multi-verifier approach with ARW does not merely boost quantitative metrics; it fundamentally elevates perceptual quality. As proven in our Human Evaluation (Fig. 5), the multi-verifier setup significantly outperforms single-verifier guidance in overall human perceived quality.

---

### Review · Reviewer_B8vC · 2026-03-13

**Summary Of Contributions:**

This paper delivers the systematic study of Inference-Time Scaling (ITS) for joint audio-video generation, tackling the challenge of extending single-modality ITS to multimodal scenarios with heterogeneous optimization objectives. It identifies the critical flaws of single-verifier guidance and validates the necessity of a multi-verifier framework for holistic quality improvement, further pinpointing the optimal combination of VideoReward-TA and JavisScore . The paper also proposes Adaptive Reward Weighting (ARW), a novel test-time optimization algorithm that adaptively calibrates heterogeneous reward variances via learnable parameters, eliminating the need for prior knowledge of reward distributions and preventing single reward dominance in aggregation. Additionally, extensive experiments on the VGGSound and JavisBench-mini benchmarks with the JavisDiT and MMDisCo models, together with human evaluations and ablation studies, fully demonstrate the proposed framework’s significant improvements in the semantic alignment, perceptual quality and audio-visual synchronization of generated audio-video pairs; the work also analyzes the compute-performance trade-offs of the method, discusses its inherent limitations and outlines promising future research directions for ITS in multimodal generation.

**Audience:**

Yes

**Audience Explanation:**

- The multi-verifier framework and ARW algorithm are not limited to joint audio-video generation; their core ideas are directly transferable to other multimodal generation tasks and even single-modality tasks with multiple evaluation objectives. This makes the work a valuable reference for researchers working on inference-time optimization of generative models across domains.

- Joint audio-video generation is a key enabler for applications such as film production, game content creation, AR/VR, and multimedia content synthesis. The proposed training-free ITS framework improves output quality without the need for costly retraining or larger models, lowering the barrier to entry for researchers and practitioners with limited computational resources—an important consideration for the broader ML community.

- The paper provides systematic empirical insights into verifier selection, reward aggregation, and compute-performance trade-offs for multimodal ITS. These insights inform future research on scaling inference-time compute for generative models, a rapidly growing area of interest in ML.

- The paper demonstrates how to effectively manage multi-objective optimization in test-time for high-dimensional multimodal data, a problem that plagues many modern generative AI systems. The results provide a blueprint for designing robust test-time optimization pipelines for other multi-objective generative tasks.

**Broader Impact Concerns:**

N/A.

**Claims And Evidence:**

Yes

**Claims Explanation:**

- The paper tests the proposed framework on two distinct benchmarks and two state-of-the-art models, reporting performance across a diverse set of metrics that cover text-modal alignment, audio-visual consistency, temporal synchronization, and video perceptual quality. Direct comparisons between single/multi-verifier guidance, different verifier combinations, and various reward aggregation methods (Tables 1–6) provide clear numerical evidence for the superiority of the multi-verifier framework and ARW over baselines.

- The paper conducts targeted ablation analyses on critical components, including audio-video synchronization verifiers (Table 10), text-video consistency verifiers (Table 7), preference weight sensitivity of ARW, and hyperparameter tuning for weighted sum aggregation (Table 11). These studies isolate the impact of each component and validate the design choices of the framework, ruling out alternative explanations for the observed performance gains.

- Qualitative results (Figs. 1, 6, 8–10) visually demonstrate the framework’s ability to capture fine-grained semantic and temporal details that naive sampling and single-verifier guidance fail to generate. Human evaluations (Fig. 5) further validate that the method aligns with human preferences for text consistency, audio-visual alignment, and overall quality, addressing the potential gap between automatic metrics and human perception.

- The paper quantifies the computational overhead of ITS (wall-clock time, number of function evaluations) and its correlation with performance gains (Table 9), providing a realistic assessment of the method’s practicality—an important consideration for validating claims about the utility of the proposed framework.

**Requested Changes:**

- The paper mentions that base model fidelity limits the performance of ITS, but it does not provide specific examples of failure cases. Adding a small section with failure case visualization and analysis (e.g., complex prompts, rare events, fine-grained temporal details) would help the community understand the current limitations of the approach and guide future work on improving base models for joint audio-video generation.

- While the paper describes the ARW algorithm (Algorithm 1), it does not report the convergence behavior of the ARW optimization (e.g., how many iterations are needed for the learnable scale parameters to stabilize, or the impact of different optimizers on ARW performance). Adding a small figure/table in the appendix showing convergence curves for ARW would improve the interpretability of the algorithm and help practitioners implement it effectively.

- The paper compares the compute cost of different ITS strategies but does not provide a breakdown of the computational overhead of individual verifiers (e.g., VideoReward-TA vs. JavisScore vs. AVHScore). This information is critical for practitioners deploying the framework in real-world settings, as it allows them to make trade-offs between verifier performance and inference speed. Adding a short table with verifier inference time/GFLOPs would address this gap.

- The paper validates ARW for joint audio-video generation; a brief experimental analysis of ARW’s performance on a simpler multimodal task would demonstrate the algorithm’s generalizability and strengthen its claim as a universal reward aggregation method for test-time optimization. If experimental resources are limited, a conceptual discussion of how ARW could be adapted to other tasks is sufficient.

- The paper tests the framework on standard benchmarks but does not evaluate its performance on diverse prompt types . Adding a small section with results on prompt variability would demonstrate the framework’s robustness and make it more valuable for real-world use cases where prompts are often unstructured.

- The paper compares ARW with standard baselines but does not discuss its relationship to state-of-the-art multi-objective optimization methods (e.g., Pareto-based methods, uncertainty-aware multi-task learning beyond Kendall et al. 2018). Adding a short discussion in the related work/conclusion section that contrasts ARW with these methods and highlights its unique advantages would strengthen the paper’s theoretical positioning.

- The paper notes that ITS incurs substantial computational overhead due to generating/evaluating multiple candidates. Adding a brief discussion of potential efficient search strategies would align the paper with its future work agenda and provide actionable guidance for practitioners looking to deploy the framework at scale.

---

> ### Author Response · Authors · 2026-03-23
> **Response to Reviewer B8vC - Part I**
>
> We sincerely thank the reviewer for the positive and constructive feedback. We are glad that the reviewer found our main contributions clear and well supported, including the systematic study of ITS for joint audio-video generation, the necessity of the multi-verifier framework, the effectiveness of ARW, and the extensive empirical validation across benchmarks, models, human evaluation, and ablations.
>
> Following the reviewer’s helpful suggestions, we further strengthened the paper with additional experiments, qualitative analysis, and discussion. We respond to each requested change below.
>
> >Q1. A failure-case analysis would strengthen the paper.
>
> **Ans:** We thank the reviewer for this helpful suggestion. In the revised manuscript, we added **Appendix H** and **Figure 16** to show failure cases that ITS cannot address perfectly despite its superiority over naive sampling. In particular, we highlight physical implausibility (the hummingbird’s wings passing through the feeder) and unstable temporal consistency (persistent flickering). These cases further support our discussion that ITS is still bounded by the base fidelity of current joint audio-video generators. We kindly ask the reviewer to check the added Appendix H and Figure 14 to help clarify these limitations.
>
> >Q2. Reporting the convergence behavior of ARW would improve interpretability.
>
> **Ans:** In the revised manuscript, we added **Appendix D.4** and **Figure 8** to report the convergence behavior of ARW under different optimizers. Specifically, we compare Adam, SGD, and RMSprop by tracking both the calibration loss and the learned scale parameters over optimization steps.
>
> The results show that all three optimizers converge to very similar scale parameters, and the loss functions typically stabilize within roughly 50–100 steps, indicating that ARW is a straightforward and generalizable algorithm that is easy to implement.
>
> We also observe that Adam provides the smoothest convergence, while SGD and RMSprop also converge reliably after a rapid initial decrease. Based on this analysis, we keep Adam as the default optimizer in the main experiments.
>
> >Q3. A verifier-wise computational cost analysis would be valuable.
>
> **Ans:** In the revised manuscript, we added **Appendix B.3** and **Table 10** to explicitly report the computational overhead of each verifier, including the base model, number of parameters, inference time, and TFLOPs.
>
> **Table 10: Computational overhead of individual verifiers.** We report the base model, number of parameters, inference time, and theoretical compute cost (TFLOPs) for each verifier.
>
> | Category | Verifier | Base model | # Parameters | Inference time (s) | TFLOPs |
> |---|---|---|---:|---:|---:|
> | Text-consistency | VideoReward | Qwen2-VL-2B | 2B | 0.13 | 4.81 |
> | Text-consistency | VQAScore | CLIP-FlanT5-XXL | 11B | 13.73 | 857.46 |
> | AV-consistency | JavisScore | ImageBind-Huge | 1.2B | 1.26 | 33.69 |
> | AV-consistency | AVHScore | ImageBind-Huge | 1.2B | 1.22 | 33.10 |
>
> >Q4. A brief discussion of ARW’s generalizability beyond joint audio-video generation.
>
> **Ans:** We completely agree that demonstrating ARW’s generalizability strengthens its claim as a universal reward aggregation method.
>
> Fundamentally, ARW is modality-agnostic. Because the algorithm operates exclusively on the scalar reward values produced by the verifiers—rather than relying on the characteristics of the modalities (e.g., pixels, waveforms, or text tokens)—it can be directly plugged into any test-time search pipeline that involves heterogeneous objectives.
>
> In Text-to-Image (T2I) generation as an example, inference-time optimization frequently struggles to balance multiple conflicting objectives, such as **Semantic Alignment** (CLIPScore [1], VQAScore [2]) and **Visual Perceptual Quality** (Human Preference Scores [3], ImageReward [4]). These verifiers produce scores with vastly different scales and variances. The existing approach [5] relies on a multi-objective weighted sum, which requires exhaustive, dataset-specific hyperparameter tuning to prevent one verifier from dominating the generation. By simply replacing our audio-visual verifiers with these T2I-specific verifiers, ARW will dynamically optimize its learnable calibration parameters to balance the variances of the CLIP and Aesthetic scores across the generated image candidates. This ensures a balanced optimization without any prior offline statistics or manual weight tuning.
>
> [1] Learning Transferable Visual Models From Natural Language Supervision
>
> [2] Evaluating Text-to-Visual Generation with Image-to-Text Generation
>
> [3] Human Preference Score v2: A Solid Benchmark for Evaluating Human Preferences of Text-to-Image Synthesis
>
> [4] ImageReward: Learning and Evaluating Human Preferences for Text-to-Image Generation
>
> [5] Test-time Alignment of Diffusion Models without Reward Over-optimization

---

> ### Author Response · Authors · 2026-03-23
> **Response to Reviewer B8vC - Part II**
>
> >Q5. Results on prompt variability would further strengthen the paper.
>
> **Ans:** In the revised manuscript, we added **Appendix E**, together with **Figures 12 and 13**, to explicitly evaluate robustness under diverse and unstructured prompt types, and we kindly invite the reviewer to check these results.
>
> The qualitative results show that naive sampling is sensitive to prompt structure, often missing fine-grained details. For example, in Figure 12, naive sampling fails to generate key details such as the visible pouring stream, ripples, and bubbles, whereas ITS more faithfully preserves these semantics across all prompt variants. In Figure 13, prompt perturbations further expose the brittleness of naive sampling: the fragment-style prompt leads to an incorrectly added orange bird, and the typo-corrupted prompt (“forst”) causes a failure to generate the intended forest background. In contrast, the proposed ITS consistently generates outputs that remain much more faithful to the textual contents across all perturbation styles.
>
> >Q6. A broader discussion of ARW in the context of multi-objective optimization would improve its theoretical positioning.
>
> **Ans:** We agree that contextualizing ARW alongside state-of-the-art multi-objective optimization (MOO) methods significantly strengthens the theoretical positioning of our work.
>
> While ARW shares the foundational goal of balancing heterogeneous objectives with methods like Pareto-based optimization and uncertainty-aware multi-task learning, there is a fundamental distinction in their application scope and computational cost.
> Existing multi-objective optimization methods are predominantly designed for the *training phase*. Pareto-based methods aim to find a common gradient descent direction to update model weights without task conflicts. Similarly, uncertainty-aware MTL (Kendall et al., 2018) dynamically weights loss functions during training based on task-specific uncertainty. These approaches require access to the model's internal parameters and involve computationally heavy gradient calculations over a training dataset.
>
> In contrast, ARW introduces an *inference-time* paradigm, without needing to update the parameters of the base generators. It conceptually adapts the intuitive principle of variance-based weighting (similar to Kendall et al.) but uniquely repurposes it for test-time search over a perfectly frozen generative model. By dynamically calibrating the scales of heterogeneous reward signals using lightweight parameters, ARW offers the unique advantage of resolving multi-objective conflicts on the fly—without requiring model retraining, or prior offline statistics. This makes it highly scalable and computationally efficient for practical deployment.
>
> Following the reviewer’s suggestion, we have added a dedicated paragraph in the **Appendix A. Related Work section**. This new discussion explicitly contrasts ARW with multi-objective optimization methods (including Pareto-based methods and uncertainty-aware MTL), highlighting its unique position as a lightweight multi-objective search algorithm for inference-time scaling.
>
> >Q7. A brief discussion of efficient search strategies would be useful.
>
> **Ans:** We agree that improving the efficiency of ITS is an important future direction for practical deployment at scale. Prior works [1,2] suggest that, as denoising progresses, partially denoised intermediate states become increasingly informative of final video quality and semantic alignment, enabling intermediate rewards to guide search before full denoising. We observed a similar tendency in our own experiments. However, our additional analysis also indicates that audio-visual synchronization and text-audio consistency are much harder to assess reliably at early denoising stages, and typically require the denoising process to proceed much further before they can be measured accurately. This points to a promising direction for efficient search: if a proxy verifier could estimate AV synchronization or related multimodal alignment signals earlier in the denoising process, low-quality noise candidates could be pruned before full generation, thereby substantially reducing the overall search cost. We added this discussion to **Appendix I** in the revised manuscript.
>
> [1] Latent reward-guided search for faster inference-time scaling in video diffusion
>
> [2] Inference-Time Text-to-Video Alignment with Diffusion Latent Beam Search

---

### Review · Reviewer_m92Z · 2026-03-24

**Summary Of Contributions:**

This paper studies how Inference-Time Scaling (ITS) can be extended from single modality generation to joint audio-video generation. It shows that single-verifier guidance leads to asymmetric trade-offs and verifier hacking, motivating the need for a multi-verifier formulation. The paper then proposes a multi-verifier framework combining text-video consistency with fine-grained audio-visual synchronization to achieve high-quality joint audio–video generation. Furthermore, it introduces Adaptive Reward Weighting (ARW), a test-time optimization algorithm for aggregating heterogeneous reward signals, ensuring robust multi-objective selection. The experiments on JavisBench-mini and VGGSound show that the approach can improve joint audio-video generation quality.

**Audience:**

Yes

**Audience Explanation:**

Inference-time scaling for multimodal generation is timely, and joint audio-video generation is an important but underexplored area. Furthermore, the observation that multi-verifier guidance is more appropriate than single-verifier guidance in this domain may provide some reference.

**Broader Impact Concerns:**

I do not have major additional broader-impact concerns.

**Claims And Evidence:**

No

**Claims Explanation:**

First, the qualitative results demonstrate clear failures in capturing fine-grained prompt details. For example, the text prompt in Figrue.1 specifies “Four birds with green bodies, red heads, and long tails,” yet the generated birds are uniformly green. These failures indicate that the proposed multi-verifier and ARW methods do not reliably enforce multi-objective constraints.

Second, the evaluation overlooks the impact of the proposed search algorithms on output diversity. It is a well-know phenomenon that applying strong reward-guided selection, such as Best-of-N or EvoSearch, forces the generative model to collapse toward specific, high-reward modes, fundamentally trading diversity for alignment.

Finally, the proposed ARW method is closely related in spirit to standard scale/variance calibration techniques.

**Requested Changes:**

Fix the qualitative evidence and thoroughly audit all showcased examples. The Figure 1 should be replaced, and the paper should ensure that headline examples actually support the stated prompt-alignment claims.

Strengthen the empirical case for ARW with stronger baselines and sharper ablations.

---

> ### Author Response · Authors · 2026-03-30
> **Response to Reviewer m92Z - Part I**
>
> We thank the reviewer for the valuable time and effort to evaluate our paper. We hope our detailed responses below can help address the reviewer's concerns.
>
> >Q1. Fix the qualitative evidence and thoroughly audit all showcased examples.
>
> **Ans:** We sincerely thank the reviewer for the thorough audit of our qualitative examples and for this constructive feedback. We completely agree that the headline example should flawlessly demonstrate the strengths of our framework.
>
> First, we would like to kindly emphasize that our extensive quantitative and qualitative results consistently demonstrate that the proposed ITS framework follows text prompts much more faithfully than naive sampling. As firmly supported by our quantitative metrics (Tables 1 and 2), human evaluation results (Figure 5), and various visual comparisons (Figures 6, 9, 10, 11, 12 and 13), ITS reliably enforces multi-objective constraints and significantly enhances prompt alignment across diverse generation scenarios.
>
> Second, we acknowledge that ITS is not entirely immune to errors. As discussed in our limitation and failure cases sections (Appendix H), the final performance of ITS is still bounded by the base fidelity of the underlying generator. If the base model fundamentally fails to generate specific attribute combinations across all candidates in the search pool, ITS can only select the best available candidate.
>
> Nevertheless, we agree with the reviewer that Figure 1 should present a cleaner and more representative headline example. Following this suggestion, we have completely replaced Figure 1 in the revised manuscript with a new example that perfectly captures all detailed prompt attributes. We kindly ask the reviewer to check the revised **Figure 1**.
>
> >Q2. Strengthen the empirical case for ARW with stronger baselines and sharper ablations.
>
> **Ans:** To further strengthen the empirical case for ARW on a stronger AV-generation baseline, we conducted an additional experiment on **LTX-2** [1], a recent open-source joint audio-video generation model. We compare naive sampling and multi-verifier ITS with ARW under the Best-of-N setting on a subset of 100 prompts from JavisBench-mini, chosen due to the substantial computational cost of ITS on this backbone.
>
> As shown in **Table 13**, ARW consistently improves over naive sampling on LTX-2, achieving +10.43% text improvement, +7.36% AV improvement, and +9.11% overall improvement. We also provide qualitative comparisons in **Figures 14 and 15**, which further show that ARW improves prompt-faithful audiovisual generation on this stronger model. We would also be grateful if the reviewer could kindly check the additional qualitative results in Figures 14 and 15, as well as the anonymized demo page linked here: **[[anonymous demo link](https://anonymous-avits.github.io/anonymous_ltx2/)]**.
>
> We believe this additional result meaningfully strengthens the empirical case for ARW. Our original paper already demonstrated gains on two distinct AV-generation models, JavisDiT and MMDisCo, showing that the method is not tied to a single architecture. The new LTX-2 experiment further extends this evidence to a stronger public AV-generation model, reinforcing that ARW is a backbone-agnostic inference-time framework rather than a model-specific technique.
>
> **Table 13: Performance of LTX-2 using ITS.**
>
> | Aggregation | VR | VQA | TV-IB | TA-IB | AV-IB | AVH-Score | JavisScore | Text | AV | Overall |
> |---|---:|---:|---:|---:|---:|---:|---:|---:|---:|---:|
> | Naive sampling | 0.515 | 0.912 | 0.264 | 0.165 | 0.260 | 0.254 | 0.226 | – | – | – |
> | ARW (Ours) | 0.683 | 0.923 | 0.264 | 0.178 | 0.278 | 0.270 | 0.246 | **10.43%** | **7.36%** | **9.11%** |
>
> [1] LTX-2: Efficient Joint Audio-Visual Foundation Model

---

> ### Author Response · Authors · 2026-03-30
> **Response to Reviewer m92Z - Part II**
>
> >Q3. The evaluation overlooks the impact of the proposed search algorithms on output diversity. It is a well-know phenomenon that applying strong reward-guided selection, such as Best-of-N or EvoSearch, forces the generative model to collapse toward specific, high-reward modes, fundamentally trading diversity for alignment.
>
> **Ans:** We agree that Best-of-N can restrict diversity, since it simply selects the highest-reward sample from a fixed candidate set and does not actively explore beyond the initial pool. However, we would respectfully clarify that this concern does not apply uniformly to all ITS methods. In particular, recent work on EvoSearch [1] was explicitly motivated by the limitation that prior reward-guided search methods can suffer from reward over-optimization and reduced diversity, and it introduces selection and mutation mechanisms designed to preserve population diversity while continuing to explore new states. The paper further reports that EvoSearch achieves higher diversity than both Best-of-N and particle sampling while also obtaining higher reward, rather than collapsing more strongly to a single high-reward mode.
>
> More importantly, in our setting, the primary goal of ITS is to identify a high-quality sample that best matches the user’s text prompt while also achieving audio-video alignment. In joint audio-video generation, both prompt-faithful semantic consistency and cross-modal alignment are central objectives of candidate selection, whereas unconstrained variation induced purely by random noise is less useful if it does not improve either semantic fidelity or audiovisual coherence. From this perspective, some alignment–diversity trade-off can exist in reward-guided search, but the key question in our paper is whether the selected sample better satisfies the intended prompt and exhibits stronger audio-video alignment.
>
> [1] Scaling Image and Video Generation via Test-Time Evolutionary Search
>
> >Q4. The proposed ARW method is closely related in spirit to standard scale/variance calibration techniques.
>
> **Ans:** We would like to further clarify the practical effectiveness of ARW. **Table 12** shows that the Weighted Sum baseline requires manual tuning to find an appropriate weight for each reward signal, whereas ARW performs this calibration automatically at test time. While Z-score normalization is a competitive baseline, it depends on pre-computed reward statistics from a training set, which makes it less practical when such statistics are unavailable at test time and potentially more sensitive to distribution shift. In contrast, ARW achieves comparable or better performance without requiring any prior statistics. Moreover, compared with other statistic-free baselines such as Rank, Min-Max, and Weighted Sum, ARW consistently yields stronger performance improvements across benchmarks.
>
> We also clarify that ARW introduces virtually no practical latency. ARW does not train or update the generator or the verifiers. Instead, it only optimizes a small number of scalar calibration parameters using a lightweight optimizer. As shown in **Table 9**, the wall-clock overhead of ARW compared to the naive Weighted Sum baseline is only 0.06%–0.08%, which is negligible relative to the cost of diffusion sampling itself.

---

### Decision · Action_Editor_2Q4F · 2026-05-18

**Recommendation:** Accept as is

**Audience:**

Yes

**Audience Explanation:**

The topic is timely and relevant for the community. All reviewers generally agreed that extending ITS to multimodal generation is an important and underexplored direction.

**Claims And Evidence:**

Yes

**Claims Explanation:**

This paper studies inference-time scaling (ITS) for joint audio-video generation through a multi-verifier framework and Adaptive Reward Weighting (ARW) for balancing heterogeneous reward signals during test-time search. During the review process, reviewers raised concerns regarding the strength of the empirical evidence, the novelty and positioning of ARW compared to existing methods, and the practical trade-offs of the proposed framework. During the rebuttal, the authors provided clarifications and additional results that substantially strengthened the paper, addressing most concerns raised by the reviewers. The revision includes additional experiments using a different backbone model, qualitative results, additional analyses on convergence behavior, etc.

While some concerns remain, the overall consensus of the reviewers is that the paper provides a meaningful empirical and methodological contribution to inference-time optimization for multimodal generation, all claims made by the authors are well supported, and would be of interest to the community.